# Learning from a Learning User for Optimal Recommendations

## Abstract

In real-world recommendation problems, especially those with a formidably large item space, users have to gradually learn to estimate the utility of any fresh recommendations from their experience about previously consumed items. This in turn affects their interaction dynamics with the system and can invalidate previous algorithms built on the omniscient user assumption. In this paper, we formalize a model to capture such "learning users" and design an efficient system-side learning solution, coined Noise-Robust Active Ellipsoid Search (RAES), to confront the challenges brought by the non-stationary feedback from such a learning user. Interestingly, we prove that the regret of RAES deteriorates gracefully as the convergence rate of user learning becomes worse, until reaching linear regret when the user's learning fails to converge. Experiments on synthetic datasets demonstrate the strength of RAES for such a contemporaneous system-user learning problem. Our study provides a novel perspective on modeling the feedback loop in recommendation problems.

## 1. Introduction

A recommender system (hereinafter referred to as *system*) is designed to predict users' preferences over items so as to maximize the utility of the recommended items (Sarwar et al., 2001; Koren et al., 2009). Driven by this principle, there has been a tremendous amount of research efforts and industry practices on developing various recommendation algorithms that predict item utility for each user based on the observed user-item interactions, including collaborative filtering (Sarwar et al., 2001; Konstan et al., 1997; Linden et al., 2003), latent factor models (Koren et al., 2009; Rendle, 2010; Rendle & Schmidt-Thieme, 2010), neural recommen-

dation models (He et al., 2017; Ebesu et al., 2018; Liang et al., 2018), and sequential recommendation models (Kang & McAuley, 2018; Tang & Wang, 2018; Wu et al., 2020).

Nevertheless, this paradigm is built on an overly simplified user model: users are omniscient about the (millions of) items and thus allow the system to directly query their preferences. This assumption ceases to be true in real-world recommendation applications where the size of the item space could be formidably large. As a result, instead of being a static "classifier" (Das et al., 2007; Li et al., 2010; Linden et al., 2003), an ordinary user typically is also *learning* the item utility from her interactions with the system. For instance, a user might be new to a category of items; thus, her responses to such items can only be accurate after consuming the recommended items, possibly even after multiple times.

This "inaccuracy" in users' feedback cannot be simply modeled as random noise, since it naturally depend on the interaction history and thus could be biased by her previous choices. More specifically, any small bias (e.g., towards a particular item category) in the system's past recommendations will bias the user's learning, which consequently leads to biased user feedback, which then further bias the system's subsequent recommendations. This forms a vicious circle – even if an optimal item is recommended to the user, she might not take it due to her currently inaccurate utility estimation; but failing to consume the optimal item will stop the user from exploring that direction, and thus leading to repeated future rejections of the same optimal recommendations. This is similar to the explore-exploit dilemma in bandit problems, but is much worse because in bandit problems the noise of user feedback is independent from the interaction history, whereas here the bias will accumulate. Our problem setting also differs from reinforcement learning where the reward function is fixed by the environment and independent from the agent's actions.

To address the limitation caused by the previous omniscient user assumption, we propose to model a user as an autonomous agent who is learning to evaluate the utility of system's recommendations from her interaction history. We formulate the system-user interaction in a dueling bandit setup (Yue et al., 2012), such that the user does not need to explicitly disclose their estimated utility of a chosen item.

[1]Anonymous Institution, Anonymous City, Anonymous Region, Anonymous Country. Correspondence to: Anonymous Author <anon.email@domain.com>.

Preliminary work. Under review by the International Conference on Machine Learning (ICML). Do not distribute.

This more challenging feedback assumption is motivated by the observation that an ordinary user will most often take action that fulfills her information needs with the least effort, and thus does not bother providing details, e.g., numerical ratings (Tétard & Collan, 2009). Specifically, we assume at each time step, the system proposes two items for the user and can only observe the user's choice between the two items, i.e., comparative feedback. The system aims at minimizing the cumulative regret from the interaction with the user in a given period $T$.

A very important distinction from the contextual dueling bandit problem (Dudík et al., 2015) is that we assume the user does not know the best choice ahead of time and will respond to current recommendations based on learned parameters from her past experience. Our model of such a learning user is quite general, without any need of restricting to specific learning algorithms or to any user decision rules. Our only assumption about the user learning is that the user learns to evaluate new items' utility based on her consumed items, and her estimation uncertainty on an item is proportional to the projection of this item onto the consumed item space. Natural examples include a user equipped with LinUCB (Li et al., 2010) or simply using the least square estimator (LSE) over history. Our user behavior assumption also considers potentially large estimation error and accounts for different decision making pattern under uncertainty (i.e., being optimistic, pessimistic, or purely myopic), which we will elaborate in later sections.

Our contributions are twofold. First, we propose a more realistic (though challenging) problem setting for interactive recommendation. Second, we design a learning algorithm for the system, named *Noise-robust Active Ellipsoid Search* (RAES), to make efficient learning possible when dealing with a learning user. We prove RAES enjoys a regret upper bound of $\tilde{O}(d^2 T^{\frac{1}{2}+\gamma})$, which deteriorates gracefully in $\gamma$, i.e., the convergence rate of user's learning. In addition, we present a lower bound to confirm the tightness of our regret bound and present empirical studies comparing RAES with relevant baselines.

## 2. Related Work.

The first related direction is the dueling bandit problem. First proposed by Yue & Joachims (2009), dueling bandit models an online learning problem where the feedback at each step is restricted to a noisy comparison between a pair of arms. In follow-up works, Ailon et al. (2014) developed solutions by proposing a black-box reduction from dueling bandit to classic multi-armed bandit (MAB), Dudík et al. (2015) studied the adversarial and contextual extensions of dueling bandit and generalized the solution concept. Our feedback assumption is fundamentally different from that in dueling bandit as the user's feedback evolves as she learns

from the realized rewards. This coupled environment results in the failure of almost all existing dueling bandit algorithms, including those mentioned above, as we will demonstrate in our empirical study.

The ellipsoid method serves as a key building block in our algorithm design. First proposed by Grötschel et al. (1981); Karmarkar (1984), the ellipsoid method is used to prove linear programs are solvable in polynomial time. Such an elegant idea has found applications in preference elicitation (Boutilier et al., 2006), recommender systems design (Viappiani & Boutilier, 2009; Gollapudi et al., 2021), and feature-based dynamic pricing (Cohen et al., 2020; Lobel et al., 2018). The main challenge in applying the ellipsoid method to our problem is that due to the user's inaccurate feedback, the system cannot control the intersection of the cutting hyperplane and thus needs to determine when to shrink the uncertainty set adaptively.

## 3. The Problem of Contemporaneous System-User Learning

As mentioned in the introduction, our setup inherits from the celebrated contextual dueling bandit problem but considers intrinsically different user behaviors, i.e., a learning and thus dynamically evolving user. Let $\mathcal{A}$ be the set of candidate items (henceforth, the *arms*) that the system can recommend at each round $t \in [T]$. We are interested in scenarios where $\mathcal{A}$ is formidably large and diverse. Our results hold for arbitrary $\mathcal{A}$, continuous or discrete, so long as it has a non-trivial interior and is sufficiently "dense" (see formal definitions later). The user's expected utility of consuming any arm $\boldsymbol{a} \in \mathcal{A}$ is governed by a hidden preference parameter $\theta_* \in \mathbb{R}^d$ and, specifically, is realized by the linear reward function $\theta_*^\top \boldsymbol{a}$. At each round $t$, the system recommends a pair of arms $(\boldsymbol{a}_{0,t}, \boldsymbol{a}_{1,t})$ and the user chooses one of them, i.e., the comparative feedback as in dueling bandits. We assume that the user does not know $\theta_*$ either and relies on her current estimation $\theta_t$ to make a choice between $(\boldsymbol{a}_{0,t}, \boldsymbol{a}_{1,t})$. Since any non-zero scaling on $\theta_*$ does not affect the user's feedback, we assume $\|\theta_*\|_2 = 1$ without loss of generality.

The key conceptual contribution of our problem setup is a formal non-stationary user model that captures a wide range of user-system interactions yet still permits tractable analysis of online learning with non-trivial regret guarantees. We defer a formal description of this user model to Section 3.1, and only summarize the interaction protocol at each round $t \in [T]$ as follows:

1. The system recommends $(\boldsymbol{a}_{0,t}, \boldsymbol{a}_{1,t}) \in \mathcal{A}^2$ to the user.
2. The user uses $\theta_t$, i.e., her estimation of $\theta^*$ at time $t$, to choose an arm from $(\boldsymbol{a}_{0,t}, \boldsymbol{a}_{1,t})$, denoted as $\boldsymbol{a}_t$.
3. The user observes reward $r_t$ and updates $\theta_{t+1}$ based on her observed history $\mathcal{H}_t = \{(\boldsymbol{a}_s, r_s)\}_{s=1}^t$.

4. The system observes the user's choice $a_t$ and updates its recommendation policy.

The learning objective for the system is to minimize the regret defined as

$$R_T = \sum_{t=1}^{T} \theta_*^\top (2a_* - a_{0,t} - a_{1,t}), \qquad (1)$$

where $a_* = \arg\max_{a \in \mathcal{A}} \theta_*^\top a$.

Next we introduce the remaining core components of the user behavior model by specifying: 1). her method for estimating $\theta_t$; and 2). her strategy for selecting an arm based on $\theta_t$. We refer to them as the *estimation rule* and the *decision rule* respectively.

### 3.1. Modeling a Learning User

We consider a general model of a learning user as follows.

1. (Estimation Rule) The user collects the past observations $\mathcal{H}_{t-1}$ and calculate $\theta_t = F(\mathcal{H}_{t-1})$ using any learning algorithm $F$, such that

$$\|\theta_* - \theta_t\|_{V_t} \leq c_1 t^{\gamma_1} g(\delta) \qquad (2)$$

holds with probability $1 - \delta$, where $V_t = V_0 + \sum_{s=1}^{t-1} a_s a_s^\top$, $\gamma_1 \in (0, \frac{1}{2})$ and $c_1$ are constants such that $c_1$ is independent of $t$. $V_0$ is assumed to be any Positive Semi-definite (PSD) matrix that summarizes the user's *prior knowledge* regarding the item space. One can interpret $V_0$ as $\sum_{i=1}^{n} a_{-i} a_{-i}^\top$, where $a_{-i}$ is the user's consumed item before engaging with the system. The spectrum of $V_0$ thus reflects the estimation accuracy regarding different directions of the item space. For example, if $V_0$ has some small eigenvalues, the user's response can be inaccurate in the corresponding eigen-directions. Our algorithm does not depend on the exact knowledge about $V_0$, but only on a lower bound estimation of its smallest eigenvalue.

2. (Decision Rule) When facing recommendations $(a_{0,t}, a_{1,t})$, the user makes the decision based on the following *index* which combines her estimated utility and an explorative bonus term

$$\hat{r}_i = \theta_t^\top a_{i,t} + \beta_t^{(i)} \|a_{i,t}\|_{V_t^{-1}}, i = \{0, 1\}, \quad (3)$$

where $\{\beta_t^{(0)}\}_{t \in [T]}$ and $\{\beta_t^{(1)}\}_{t \in [T]}$ are two *arbitrary* sequences satisfying $\beta_t^{(i)} \in [-c_2 t^{\gamma_2}, c_2 t^{\gamma_2}]$ for some constant $c_2$ and $\gamma_2$. Then, the user returns her choice $a_t$ with the largest index $\hat{r}$ (breaking ties arbitrarily).

In essence, the estimation rule captures a crucial property of a learning user – the utility estimation for an item becomes more accurate only when the user has experienced more similar items before. This is reflected in the data-weighted matrix norm in (2). In other words, the user's response will not be reliable if the recommended item is barely related to her previously experienced items. A similar assumption is made to capture the user's explorative behaviors for previously unseen items, as described by (3). This is fundamentally different from classical recommendation settings, where the uncertainty in user feedback is modeled by homogeneous noise of the same scale throughout the course of user-system interactions.

Next we describe a learning user example, which is also the running example of our (more general) user behavior model. As the true underlying utility function is linear, i.e., $r_t = \theta_*^\top a_t + \eta_t$, where $\eta_t$ is sub-Gaussian noise, linear regression is a natural choice for a learning user's estimation rule and its estimation confidence bound satisfies $\|\theta_* - \theta_t\|_{V_t} \leq O\left(\sqrt{d \log \frac{t}{\delta}}\right)$ with probability $1 - \delta$ (Lattimore & Szepesvári, 2020). In this case, $\gamma_1$ can be any positive number and $g(\delta) = \sqrt{\log \frac{1}{\delta}}$. But our user model covers more general estimation methods than linear regression. For example, to capture the scenario where an ordinary user does not necessarily have the capacity to precisely execute such a sophisticate estimation method, we allow the user's estimation to have much larger error at the order of $O(t^{\gamma_1})$ as in (2), where the parameter $\gamma_1$ controls the convergence rate of user learning.

The decision rule accounts for a user's potential exploration behavior when facing uncertainty, which has been observed and supported in many studies in cognitive science (Cohen et al., 2007; Daw et al., 2006) and behavior science (Gershman, 2018; Wilson et al., 2014). One natural option is to follow the "optimism in the face of uncertainty" (OFUL) principle (Abbasi-Yadkori et al., 2011). Specifically, if $\theta_t$ is the least square estimator, a learning user employing the celebrated LinUCB can be realized by setting $\beta_t^{(0)} = \beta_t^{(1)} = O(\sqrt{\log t})$ in (3). But our decision rule in (3) is, again, much more general. To capture cases where users use a much looser confidence bound estimation or even less rational arm choices, we allow $\beta_t^{(i)}$ to deviate in a much larger range with $O(t^{\gamma_2})$ (compared to $O(\sqrt{\log t})$ in LinUCB). Additionally, we allow $\{\beta_t^{(i)}\}_{t \in [T]}$ to be *arbitrary* and even consist of negative values. This enables us to model highly non-stationary user behaviors, e.g., being optimistic, pessimistic, purely myopic (when $\beta_t^1 = \beta_t^0 = 0$), or an arbitrary mixture of any of them.

Parameters $\{\gamma_1, \gamma_2\}$ depict the user learning's convergence rate and user's exploration strength, respectively. Notably, we are only interested in the regime $(\gamma_1, \gamma_2) \in [0, \frac{1}{2}) \times [0, \frac{1}{2})$, because trace$(V_t)$ is in the order of $O(t)$ by the definition of $V_t$. Therefore, we must have $\|\theta_* - \theta_t\|_{V_t} = O(\sqrt{t})$

whenever $\theta_*$ is within a constant $\ell_2$ distance to the user's estimated parameter $\theta_t$. As a result, if $\gamma_1 \geq 1/2$, it must be that the estimated $\theta_t$ is at least a constant distance away from the true $\theta_*$, and so is the estimated reward $\hat{r}_i$ from the expected true reward. This makes it impossible for the system to do no-regret learning. Similarly, $\|\boldsymbol{a}_{i,t}\|_{V_t^{-1}}$ will be $\Theta(\sqrt{t})$ for some $\boldsymbol{a}_{i,t}$ and a $\gamma_2 \geq 1/2$ will also make the estimated $\hat{r}_i$ arbitrarily bad. As we will demonstrate in later analysis, the estimation error of $\hat{r}$ turns out to be governed by $\max\{\gamma_1, \gamma_2\}$. Hence, for the ease of references, in the following analysis we conveniently refer to the above user behaviors as $(c, \gamma)$-*rationality*, formally defined as:

**Definition 3.1.** $[(c, \gamma)-$rationality] Any user characterized by Estimation Rule (2) and Decision Rule (3) is said to be $(c, \gamma)-$rational if $\gamma \geq \max\{\gamma_1, \gamma_2\}, c \geq \max\{c_1, c_2\}$.

As a concrete example, a user is $(c, \gamma)$-rational for an arbitrarily small $\gamma$ if she runs LinUCB [1]. This is because under LinUCB we have $\|\theta_* - \theta_t\|_{V_t} = O(\sqrt{\log t})$ and $\{\beta_t^{(0)}, \beta_t^{(1)}\}$ are also both in the order $O(\sqrt{\log t})$. Therefore, $\gamma$ here can be an arbitrarily small positive number since $\frac{\log t}{t^\gamma} \to 0$ as $t \to \infty$ for any $\gamma > 0$.

# 4. No-Regret System Learning from a Learning User

In this section, we develop an efficient learning algorithm for the system to learn from *any* $(c, \gamma)$-rational user. The regret of our algorithm has an order of $\tilde{O}(cd^2 T^{\frac{1}{2}+\gamma})$. Recall that, a user using the LinUCB algorithm corresponds to an arbitrarily small $\gamma$. In this case, system learning essentially recovers the optimal $O(\sqrt{T})$ regret in bandit learning, despite that the system (1) only has limited comparative feedback about the user's utility estimation; and (2) faces non-stationary and non-stochastic user behaviors. More interestingly, our algorithm's regret deteriorates gracefully as $\gamma \in [0, \frac{1}{2})$ increases, i.e., as the user's learning converges at a slower rate or being more explorative as captured by $\gamma$. The key conceptual message from our theoretical findings is that it is possible for a system to learn from a learning user, and *the convergence rate of the system's learning deteriorates linearly in the convergence rate of the user's learning.*

The only caveat for our analysis is the $O(d^2)$ dependence in the regret upper bound, which is worse than the regret's linear dependence on $d$ for standard no-regret learning problems. We believe this worse dependence is fundamentally due to the fact that the system has to learn from the users' binary feedback with diminishing yet *non-stochastic* noise. This more challenging setup invalidates classic linear contextual bandit algorithms that rely on rewards with stochastic

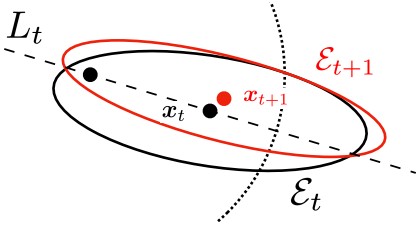

**Figure 1.** The unit ball (dashed line) is centered at the origin. $L_t$ crosses the origin, cuts $\mathcal{E}_t$ through its center $\boldsymbol{x}_t$ and yields $\mathcal{E}_{t+1}$. In a high dimensional space, we have additional degree of freedom to pick $L_t$ that shrinks $\mathcal{E}_t$ along all possible directions.

noise. We thus develop an entirely different solution, which is a novel use of the celebrated *ellipsoid method* originally developed for solving linear programs (necessary technical details of the ellipsoid method are provided in Appendix A for curious readers) (Grötschel et al., 1993; 1981). Our idea is to maintain a sequence of confidence ellipsoid $\{\mathcal{E}_t\}$ for $\theta_*$ and reduce the volume of $\mathcal{E}_t$ via a carefully chosen cutting hyperplane. The user's binary comparative feedback then tells which side of the hyperplane contains the true parameter, which prepares the subsequent cuts.

## 4.1. Warm-up: Fast Learning from a Perfect User

**A (significantly) simplified setup.** To illustrate the main idea of our solution, we start with a stylized situation, where we make the following simplifications: 1). the user knows $\theta_*$ precisely and makes decisions by directly comparing $\theta_*^\top \boldsymbol{a}_{0,t}$ and $\theta_*^\top \boldsymbol{a}_{1,t}$; 2). the action set is simply the unit ball $\mathcal{A} = \{\boldsymbol{a} : \|\boldsymbol{a}\|_2 \leq 1\}$.

**Technical Highlight I: Novel Use of the *Ellipsoid Method*.** Algorithm 1 describes our solution under this simplified problem setting. We should note Algorithm 1 differs from the classic ellipsoid method in two aspects. First, our algorithm has the freedom to *actively* choose the hyperplane $L_t$ by picking $\{\boldsymbol{a}_{0,t}, \boldsymbol{a}_{1,t}\}$ (thus named "Active Ellipsoid Search"), while the classic ellipsoid method is always passively fed with an arbitrary separating hyperplane. Second, $L_t$ has to cross the origin by construction. Therefore, to accelerate the shrinkage of the volume of $\mathcal{E}_t$ (i.e., $\text{Vol}(\mathcal{E}_t)$), we prefer a cutting direction $\boldsymbol{g}_t = \boldsymbol{a}_{0,t} - \boldsymbol{a}_{1,t}$ such that $L_t$ goes through the center $\boldsymbol{x}_t$, i.e., $\boldsymbol{g}_t^\top \boldsymbol{x}_t = 0$, and $\text{Vol}(\mathcal{E}_t)$ is halved after each iteration, as illustrated in Figure 1.

Though given more freedom, we also face a strictly harder problem. Specifically, when solving LPs, it suffices to reach an ellipsoid $\mathcal{E}_t$ with a small volume where the LP *objective* is guaranteed to be approximately optimal. However, our goal here is to identify the *direction* of $\theta_*$ with small error, and thus a small $\text{Vol}(\mathcal{E}_t)$ is necessary but *not* sufficient. For instance, a zero-volume ellipsoid in $\mathbb{R}^d$ can still enclose a $d-1$ dimensional subspace and thus contains a very diverse set of directions that are far from $\theta_*$.

---

[1] This is also the reason for our terminology "rationality". That is, there exists (essentially) 0-rational learning users, so a $\gamma$-rational user for some $\gamma > 0$ must not be perfectly rational.

**Algorithm 1** Active Ellipsoid Search on Unit Sphere

1: **Input:** Dimension $d > 0$, number of iterations $T > 0$.
2: **Initialization:** $\boldsymbol{x}_0 = \boldsymbol{0}, P_0 = I_d$.
3: **while** $0 \leq t \leq T$ **do**
4:     Compute eigen-decomposition

$$P_t = \sum_{i=1}^{d} \sigma_i^{(t)} \boldsymbol{u}_i^{(t)} \boldsymbol{u}_i^{(t)\top}, \sigma_1^{(t)} \geq \cdots \geq \sigma_d^{(t)};$$

5:     Compute any unit vector $\boldsymbol{g}_t \in \text{span}\{\boldsymbol{u}_1^{(t)}, \boldsymbol{u}_2^{(t)}\}$ that is orthogonal to $\boldsymbol{x}_t$;
6:     Pick $(\boldsymbol{a}_{0,t}, \boldsymbol{a}_{1,t}) = (-\boldsymbol{g}_t, \boldsymbol{g}_t)$; and observe the user's choice $\boldsymbol{a}_{i,t}, i \in \{0, 1\}$.
7:     Set $\tilde{\boldsymbol{g}}_t = (2i - 1)\boldsymbol{g}_t / \|\boldsymbol{g}_t\|_{P_t}$;
8:     Update

$$\boldsymbol{x}_{t+1} = \boldsymbol{x}_t - \frac{1}{d+1} P_t \tilde{\boldsymbol{g}}_t;$$

$$P_{t+1} = \frac{d^2}{d^2 - 1} \left( P_t - \frac{2}{d+1} P_t \tilde{\boldsymbol{g}}_t \tilde{\boldsymbol{g}}_t^\top P_t \right).$$

9: **end while**
10: **Output:** The estimation of $\theta_*$ : $\hat{\theta}_T = \boldsymbol{u}_1^{(T)}$.

To achieve this strictly harder objective, we need $L_t$ to cut $\mathcal{E}_t$ along the direction in which $\mathcal{E}_t$ has the largest width, i.e., the most uncertain direction. This requires $\boldsymbol{g}_t$ to be aligned with the eigenvector corresponding to the largest eigenvalue of $P_t$, which is in general not compatible with $\boldsymbol{g}_t^\top \boldsymbol{x}_t = 0$. Here then comes the crux of our approach – we relax the second condition by picking $\boldsymbol{g}_t$ from a two-dimensional space spanned by the eigenvectors corresponding to the top-2 largest eigenvalues of $P_t$. Under this choice of $\boldsymbol{g}_t$, $\mathcal{E}_t$ is guaranteed to converge to a skinny-shaped ellipsoid with its longest axis converging to the direction of $\theta_*$ at an exponential rate. The detail is presented in Algorithm 1, and the convergence analysis of Algorithm 1 is formalized in the following theorem.

**Theorem 4.1.** *At each time step $t$ in Algorithm 1, let the eigenvalues of $P_t$ be $\sigma_1^{(t)} \geq \cdots \geq \sigma_d^{(t)}$. For any $d > 1, T > 0$, we have*

*1. for any $2 \leq i \leq d$,*

$$\sigma_i^{(T)} \leq \exp\left(\frac{4}{d} - \frac{T}{d^2}\right), \tag{4}$$

*2. the $\ell_2$ estimation error for $\theta_*$ is given by*

$$\left\|\theta_* - \hat{\theta}_T\right\|_2 \leq 2\sqrt{d-1} \exp\left(\frac{2}{d} - \frac{T}{2d^2}\right). \tag{5}$$

We postpone the proof of Theorem 4.1 to Appendix B. This theorem indicates that the $\ell_2$ estimation error for $\theta_*$ converges to zero at the rate of $O\left(d^{\frac{1}{2}} \exp\left(-\frac{T}{2d^2}\right)\right)$. In other

words, to guarantee $\|\theta_* - \hat{\theta}_T\|_2 < \epsilon$, at most $O(d^2 \log \frac{d}{\epsilon})$ iterations are needed.

### 4.2. Robust Learning from a Learning User

The previous section illustrates our system learning principle, but under a greatly simplified setting with a perfect user. In this section, we extend the solution to account for a learning user who does not know $\theta_*$ and keeps refining her estimation $\theta_t$. Here, the user's feedback still provides a linear inequality regarding $\theta^*$ and thus similarly serves as a cutting hyperplane. But since the user acts based on the *index* $\hat{r}_i = \theta_t^\top \boldsymbol{a}_{i,t} + \beta_t^{(i)} \|\boldsymbol{a}_{i,t}\|_{V_t^{-1}}$, the cutting hyperplane now has the form $L_t = \{\boldsymbol{z} : \boldsymbol{z}^\top(\boldsymbol{a}_{0,t} - \boldsymbol{a}_{1,t}) = \beta_t^{(1)} \|\boldsymbol{a}_{1,t}\|_{V_t^{-1}} - \beta_t^{(0)} \|\boldsymbol{a}_{0,t}\|_{V_t^{-1}}\}$. Importantly, the intercept term now depends on $\{\beta_t^{(0)}, \beta_t^{(1)}\}$ which are arbitrary within the uncertainty region $[-ct^\gamma, ct^\gamma]$.

**Technical Highlight II: Ellipsoid Search with Noise.** Due to the aforementioned noise in the users' binary feedback, we thus face an interesting challenge – how to perform the ellipsoid search under (non-stochastic) noisy feedback? Somewhat surprisingly, this basic question was not addressed in literature about ellipsoid method. We tackle this challenge by refining the ellipsoid method to tolerate carefully chosen scales of noise and decreasing the tolerance as the ellipsoid shrinks. In order to elicit more accurate feedback, our algorithm must ensure the diversity of the recommended items to prepare the user for improved precision of her responses in all directions. To this end, we improve Algorithm 1 by adaptively preparing the user until a desirable level of accuracy of her estimated $\theta_t$ is reached and then cut the ellipsoid. To our knowledge, this noise-robust version of ellipsoid method is novel by itself and may be of independent interest. We coin this new algorithm "Noise-robust Active Ellipsoid Search", or RAES in short.

**Regularity assumptions on the action set.** Before introducing the RAES algorithm, we first pose several natural and technical assumptions regarding the action set $\mathcal{A} \subset \mathbb{R}^d$. Specifically, $\mathbb{B}_p^d(0, r)$ denotes the $d$-dimensional $\ell_p$ ball centered at the origin with radius $r$. Without loss of generality, we assume $\boldsymbol{0} \in \mathcal{A} \subset \mathbb{B}_2^d(0, D_1)$ since one can always shift all actions by the same amount and then re-scale the actions without changing the users' responses.

The first assumption is a familiar one, as also used in previous works such as (Rusmevichientong & Tsitsiklis, 2010).

**Assumption 4.2** ($L$-Smooth Best Arm Response Condition, $L$-SRC)**. Let $\boldsymbol{x}_\mathcal{A}^* = \arg\max_{\boldsymbol{x}' \in \mathcal{A}} \boldsymbol{x}^\top \boldsymbol{x}', \forall \boldsymbol{x} \in \mathcal{A}$. There exists a constant $L > 0$ such that for any pair of non-zero unit vectors $\boldsymbol{x}, \boldsymbol{y} \in \mathbb{R}^d$, we have*

$$\|\boldsymbol{x}_\mathcal{A}^* - \boldsymbol{y}_\mathcal{A}^*\|_2 \leq L \cdot \|\boldsymbol{x} - \boldsymbol{y}\|_2.$$

A compact set $\mathcal{A}$ satisfies $L$-SRC if and only if $\mathcal{A}$ can be represented as the intersection of closed balls of radius $L$. Intuitively, the $L$-SRC condition requires the boundary of $\mathcal{A}$ to have a curvature that is bounded below by a positive constant. For instance, the unit ball satisfies 1-SRC, and an ellipsoid of the form $\{\boldsymbol{u} \in \mathbb{R}^d : \boldsymbol{u}^\top P^{-1} \boldsymbol{u} \leq 1\}$, where $P$ is a PSD matrix, satisfies the $\frac{\lambda_{\max}(P)}{\sqrt{\lambda_{\min}(P)}}$-SRC.

**Assumption 4.3** ($\epsilon$-Dense Condition, $\epsilon$-DC)**.** $\mathcal{A}$ is an $\epsilon$-cover of a continuous set $\bar{\mathcal{A}}$, i.e., $\bar{\mathcal{A}} \subset \cup_{\boldsymbol{x} \in \mathcal{A}} \mathbb{B}_2^d(\boldsymbol{x}, \epsilon)$. In addition, there exists constants $D_1 > D_0 > 0$ such that $\mathbb{B}_2^d(0, D_0) \subseteq \mathcal{A}, \bar{\mathcal{A}} \subseteq \mathbb{B}_2^d(0, D_1)$.

This assumption suggests the action set $\mathcal{A}$ is sufficiently dense. A continuous $\mathcal{A}$ is 0-DC. However, $\epsilon$-DC relaxes the continuity requirement on $\mathcal{A}$ by allowing $\mathcal{A}$ to take the form of an $\epsilon$-net of a continuous set $\bar{\mathcal{A}}$. For convenience of references, we associate any element $\bar{\boldsymbol{a}} \in \bar{\mathcal{A}}$ with an element $\boldsymbol{a} \in \mathcal{A}$ such that $\|\boldsymbol{a} - \bar{\boldsymbol{a}}\|_2 \leq \epsilon$. For our analysis, this relation does not need to be exclusive or reversible.

As indicated in the initialization of Algorithm 2, RAES does not rely on the exact values of $(c, \gamma, V_0)$, which could be difficult to attain in reality. Instead, any reasonable upper bounds for $c$ and $\gamma$, and a lower bound of $\lambda_{\min}(V_0)$ suffice. Similar to Algorithm 1, RAES also maintains a sequence of confidence ellipsoids $\{\mathcal{E}_t\}$. A hyper-parameter $T_0$ separates the time horizon $T$ into two phases. At time step $t$, the system first proposes the most promising cutting direction $\boldsymbol{g}_t$. However, different from Algorithm 1 which always cuts $\mathcal{E}_t$ immediately, RAES needs to compute the cutting depth $\alpha_t$ (defined in (6)) and determine whether the user's feedback is precise enough for the system to yield an improved estimation. Intuitively, $\alpha_t$ measures the normalized signed distance between the center of $\mathcal{E}_t$ and the cutting hyperplane $L_t$: $\alpha_t \in (-\frac{1}{d}, 0)$ corresponds to a shallow-cut where $L_t$ removes less than half of the volume of the ellipsoid; $\alpha_t \in (0, 1)$ corresponds to a deep-cut where more than half of the volume is reduced; and $\alpha_t = 0$ happens only when $L_t$ cuts $\mathcal{E}_t$ through the center. Since we need to deal with the uncertainty in the user's response, we may only expect shallow-cuts. Depending on $\alpha_t$ and $T_0$, the system makes a decision among the following three options, which we refer to as *cut*, *exploration*, and *exploitation*:

1. (Cut) If $t \leq T_0$ and $\alpha_t \geq -\frac{1}{kd}$, cut $\mathcal{E}_t$ and update $(\boldsymbol{x}_t, P_t)$.

2. (Exploration) If $t \leq T_0$ and $\alpha_t < -\frac{1}{kd}$, make recommendations to ensure the user is exposed to the least explored directions in $V_t$.

3. (Exploitation) If $t > T_0$, recommend the empirically best arm to the user.

The purpose of an exploration step is to prepare the user

such that a smaller $\alpha$ can be expected in the future. By the definition of $\alpha_t$, the only way to decrease it is by increasing $\lambda_{\min}(V_t)$, which can be achieved by presenting the least exposed direction to the user [2]. Finally, when the system believes the user's estimation error of $\theta_*$ is acceptable to induce a small regret, it stops preparing the user and recommends the empirically best arm when no further cut is available. The algorithm can be understood as a phase of exploration of length $T_0$ followed by a phase of exploitation, with a sequence of cut steps scattered within. The sublinear regret can be guaranteed by carefully choosing $T_0$.

Before analyzing RAES, we provide an intuitive explanation for it. First of all, the cutting direction $\boldsymbol{g}_t$ is the same as the choice in Algorithm 1, which ensures the separation hyperplane can intersect $\mathcal{E}_t$ along the most uncertain direction. Next, we translate the user's comparative feedback regarding $\theta_t$ into an inequality regarding $\theta_*$ with high probability, i.e., $\theta_*^\top \boldsymbol{g}_t \leq$ (or $\geq$)$b$, by pinning down the intersection term $b$. This can be realized by leveraging the property of the user's estimation and decision rules, resulting in the explicit form of $\alpha_t$. To simplify the technical analysis, with a slight abuse of notation, we use the subscript $t$ in $\{(\boldsymbol{x}_t, P_t)\}_{t=1}^N$ to describe the confidence ellipsoids after the $t$-th *cut* in RAES, and $N$ is the total number of cuts in horizon $T$. Lemma 4.4 characterizes the effect from each cut, exploration, and exploitation step:

**Lemma 4.4.** *If we choose*

$$\alpha_t = \tag{6}$$
$$-\frac{ct^\gamma\left(\|\boldsymbol{a}_{0,t}\|_{V_t^{-1}} + \|\boldsymbol{a}_{1,t}\|_{V_t^{-1}} + g(\delta)\|\boldsymbol{a}_{0,t} - \boldsymbol{a}_{1,t}\|_{V_t^{-1}}\right) + 2\epsilon_0}{\|\boldsymbol{g}_t\|_{P_t}}$$

*in Algorithm 2, we have*

1. *After each cut, $Vol(\mathcal{E}_{t+1}) \leq \exp\left(-\frac{(k-1)^2}{2k^2 d}\right) Vol(\mathcal{E}_t)$.*

2. *If at least $d$ exploration steps are taken starting from any time step $t_0$ to $t_0 + n$, we have $\lambda_{\min}(V_{n+t_0}) \geq \lambda_{\min}(V_{t_0}) + \frac{4D_0}{25} - 3\epsilon_0$.*

3. *At any exploitation step $t$, the instantaneous regret is upper bounded by $2L\|\theta_* - \boldsymbol{u}_1^{(t)}\|_2^2$.*

Using Lemma 4.4, we can derive the convergence rate of $\sigma_i^{(t)}$ and the regret upper bound of RAES in the following Theorem 4.5, whose proof can be found in Appendix C.

**Theorem 4.5.** *For any $d > 1, n > 0$, let $\sigma_i^{(n)}$ be the $i$-th largest eigenvalue of $P_n$ after the $n$-th cut, we have*

---

[2] A straightforward way for increasing $\lambda_{\min}(V_t)$ is to feed the user with the eigenvector corresponding to $\lambda_{\min}(V_t)$. However, to avoid forcing a user to choose between two identical items (if they are not optimal), we let the system recommend two different items.

1. For any $2 \leq i \leq d$,

$$\sigma_i^{(n)} \leq \exp\left(\frac{4}{d} - \frac{(k-1)^2 n}{k^2 d^2}\right). \quad (7)$$

2. When $T_0 = O\left(cL^{\frac{1}{2}} D_1^{\frac{1}{2}} D_0^{-\frac{3}{2}} g(\delta) d^2 T^{\frac{1}{2}+\gamma}\right)$ and $\epsilon_0 < O\left(cD_1 D_0^{-\frac{1}{2}} d^{-\frac{1}{2}} T^{-\frac{1}{4}+\frac{\gamma}{2}}\right)$, the regret of RAES is upper bounded by $O\left(cL^{\frac{1}{2}} D_1^{\frac{3}{2}} D_0^{-\frac{3}{2}} g(\frac{\delta}{T_0}) d^2 T^{\frac{1}{2}+\gamma}\right)$ with probability $1 - \delta$.

Theorem 4.5 suggests when $\mathcal{A}$ is continuous or sufficiently dense, RAES achieves a regret upper bound $\tilde{O}(cd^2 T^{\frac{1}{2}+\gamma})$ when $g(\frac{\delta}{T_0})$ grows logarithmically in $T_0$. Recall that $\gamma \in [0, \frac{1}{2})$ denotes the rationality of the user: when $\gamma$ is large, the system obtains less accurate responses from the user and thus suffers from a worse regret guarantee. When $\gamma = 0$, e.g., the user executes LinUCB, we get an upper bound of the order $\tilde{O}(\sqrt{T})$, which nearly matches the lower bound, as we will show in the following section.

### 4.3. A Regret Lower Bound

We conclude this technical section by showing a regret lower bound for the system's learning. This lower bound applies for any $\gamma > 0$, and it nearly matches the above upper bound w.r.t. time horizon $T$ when $\gamma$ is close to zero. This result leaves an intriguing open question about how tight our Algorithm 2 is for general $\gamma$, i.e., for every $\gamma \in (0, 1/2)$, what is the best possible regret for the system? We remark that resolving this open question appears to require significantly different machinaries as used in current lower bound proofs for bandit algorithms since these arguments are primarily based on information theory and thus intrinsically rely on assumption of *random* noises (Lattimore & Szepesvári, 2020; Rusmevichientong & Tsitsiklis, 2010), whereas the user's feedback noise in our model is arbitrary (though also diminishing with more rounds). We thus leave this as an interesting future direction to explore.

**Theorem 4.6.** *For any $\gamma > 0$, there exists a function $T_0(d) > 0$ such that for any $d \geq 1$, $T > T_0(d)$, and any algorithm $\mathcal{G}$ that has merely access to the comparison feedback given by a rational user defined in Definition 3.1, there exists $\theta_* \in \partial\mathbb{B}_1^d$ such that the expected regret $R_T$ defined in (1) obtained by $\mathcal{G}$ satisfies*

$$R_T^{(s)}(\mathcal{G}, \theta_*) \geq \frac{\exp(-2)}{4}(d-1)\sqrt{T}. \quad (8)$$

Theorem 4.6 may appear not surprising since, intuitively, the system's learning task appears no easier than the standard stochastic linear bandit problems for which the lower bound is already $O(\sqrt{T})$ (Rusmevichientong & Tsitsiklis, 2010). However, it turns out that delivering a rigorous proof is more subtle than this intuition, and for that we have to overcome two technical challenges: 1). adapting the current minimax lower bound proof for stochastic linear bandits to the setup where the norm of $\theta_*$ is bounded away from zero; 2). constructing a black-box reduction from the system's regret to the user's regret. Due to the space limit, we defer the proof details to Appendix D.

## 5. Experiment

In this section, we study the empirical performance of RAES to validate our theoretical analysis by running simulations on synthetic datasets in comparison with several baselines.

### 5.1. Experiment Setup and Baselines

There is no direct baseline for comparison since the learning environment we studied is new. Given the linear reward and the binary comparative feedback assumptions, we take several contextual dueling bandit algorithms for comparison, including Dueling Bandit Gradient Descent (DBGD) (Yue & Joachims, 2009), Doubler (Ailon et al., 2014), and Sparring (Ailon et al., 2014; Sui et al., 2017). The configuration of baseline algorithms and the details of the simulation environment can be found in Appendix E.

### 5.2. Experiment Results

**Robustness of RAES against a learning user:** We first demonstrate the performance of RAES under $(T, T_0, d) = (10000, 1500, 5)$ against a $(1, \gamma)$-rational user with different $\gamma$ and $V_0$ in Figure 2. Additional results for richer parameter settings are reported in Appendix E. The x-axis denotes time step $t$ and y-axis denotes the accumulated regret up to the time step $t$. The left panel illustrates the performance of RAES when $\gamma = 0.1$ and $V_0 \in \{V_0(i) : 0 \leq i \leq 5\}$, where $V_0(i)$ is the diagonal matrix with $i$ diagonal entries being 1 while other $5 - i$ entries being 100. Unsurprisingly, RAES achieves the best performance when the user has the most informative prior $V_0(0)$. When $V_0$ has small eigenvalues, RAES needs more exploration steps in the first $T_0$ rounds, but the resulting added regret is not significant. The right panel shows the result when $V_0 = I_d$ and $\gamma \in \{0, 0.1, 0.2, 0.3\}$ which confirms our theoretical analysis that the regret of RAES grows in order $O(T^{\frac{1}{2}+\gamma})$.

**Comparison with baseline algorithms:** The comparison between RAES and the three baselines against learning users are shown in Figure 3, where the x-axis denotes different time horizons $T$, and the y-axis denotes the corresponding accumulated regret. $\{\gamma, T_0\}$ are set to 0 and $0.25 \times d^2\sqrt{T}$. Additional results under different choices of $\gamma$ can be found in Appendix E. The left panel shows the result with $V_0 = 100I_d$, i.e., each algorithm is facing a well-prepared user, while the right panel is plotted with

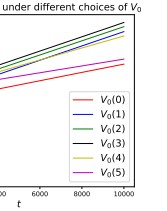
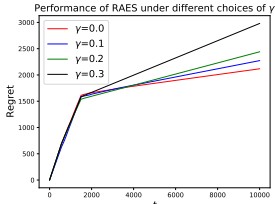

*Figure 2.* The regret of RAES against a learning user with different $V_0$ and $\gamma$ over time. Left: Fix $\gamma = 0.1$, plot for different choices of $V_0$; Right: Fix $V_0 = I_d$, plot for different choices of $\gamma$.

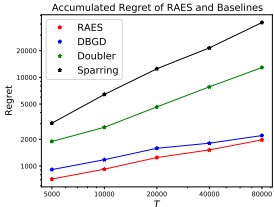
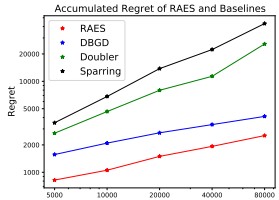

*Figure 3.* The accumulated regret of RAES and three baseline algorithms. Different colors specify different algorithms. Each star represents the accumulated regret (y-axis) of the algorithm given time horizon $T$ (x-axis) with $\gamma = 0$. Left: $V_0 = 100I_d$; right: $V_0 = \text{diag}(100, 10, 5, 2, 1)$.

$V_0 = \text{diag}(100, 20, 5, 2, 1)$. The result demonstrates that RAES enjoys the best performance and is robust against different types of learning users. Since Doubler and Sparring employ a black-box linear bandit algorithm as their subroutine, the violation of the stochastic reward assumption breaks down the linear bandit algorithm and thus the failure of the algorithms themselves. For DBGD, the left panel suggests that it can still enjoy a sub-linear regret under milder users' rationality assumptions. However, when the user's prior $V_0$ is ill-posed (i.e., $\lambda_{\min}(V_0)$ is small), the performance of DBGD deteriorates seriously. In particular, under an ill-posed $V_0$, the user's feedback can be misleading along certain directions, and the design of DBGD does not provide any mechanism to increase the accuracy of user feedback along these directions. The degradation of DBGD becomes even more evident when $\gamma$ is larger, as shown by the stark contrast in Figure 4 in Appendix E.

## 6. Conclusion

Motivated by the observation that users' feedback can be coupled with their interaction history with a recommender system, we propose a new problem setting where the system learns from non-stationary feedback of a learning user. Extending the dueling bandit framework, we formulate the problem of "learning from a learner" and establish an efficient learning algorithm based on the ellipsoid method with a near-optimal regret guarantee. Besides the new algorithm, our user learning model also provides a new perspective to studying the feedback loop in recommender systems. The negative empirical results of baseline algorithms demon-

strate how inaccuracy of user feedback is formed and amplified on the system's side in its subsequent recommendations, if failing to consider the progression of user learning. A key insight of our proposed solution is that a healthy recommender system needs to expose a diversified spectrum of items to its users and thus "foster" them to respond with informed feedback. This leads to the win-win outcome for both users and the system in exploring the item space.

---

**Algorithm 2** Noise-robust Active Ellipsoid Search (RAES)

1: **Input:** Action set $\mathcal{A} \subset \mathbb{R}^d$ with constants $(D_1, D_0, L, \epsilon_0)$, time horizon $T_0$ and $T$, cutting threshold $k > 1$, and probability threshold $\delta > 0$

2: **Initialization:** A user who is $(c, \gamma)-$rational, $\lambda_0 > 0$ be any lower bound estimation of the minimum eigenvalue of $V_0$, set $V_0 = \lambda_0 I_d$, $\boldsymbol{x}_0 = \boldsymbol{0}$, $P_0 = I_d$.

3: **while** $0 \leq t \leq T$ **do**

4:     Compute eigen-decomposition
    $P_t = \sum_{i=1}^d \sigma_i^{(t)} \boldsymbol{u}_i^{(t)} \boldsymbol{u}_i^{(t)\top}, \sigma_1^{(t)} \geq \cdots \geq \sigma_d^{(t)}$.

5:     Compute a unit vector $\boldsymbol{g}_t \in \text{span}\{\boldsymbol{u}_1^{(t)}, \boldsymbol{u}_2^{(t)}\}$ that is orthogonal to $\boldsymbol{x}_t$;

6:     Pick any pair $(\bar{\boldsymbol{a}}_{0,t}, \bar{\boldsymbol{a}}_{1,t})$ such that $\bar{\boldsymbol{a}}_{1,t} - \bar{\boldsymbol{a}}_{0,t} = m\boldsymbol{g}_t$, $m \geq 2D_0$, and compute $\alpha_t$ according to (6);

7:     **if** $t \leq T_0$ and $\alpha_t \geq -\frac{1}{kd}$ **then**

8:         Recommend $(\boldsymbol{a}_{0,t}, \boldsymbol{a}_{1,t})$, observe the user's choice $\boldsymbol{a}_t = \boldsymbol{a}_{i,t}, i \in \{0, 1\}$;

9:         Set $\tilde{\boldsymbol{g}}_t = (2i - 1)\boldsymbol{g}_t / \|\boldsymbol{g}_t\|_{P_t}$;

10:        Update

$$\boldsymbol{x}_{t+1} = \boldsymbol{x}_t - \frac{1 + d\alpha_t}{d + 1} P_t \tilde{\boldsymbol{g}}_t; \quad (9)$$

$$P_{t+1} = \frac{d^2(1 - \alpha_t^2)}{d^2 - 1}\left(P_t - \frac{2(1 + d\alpha_t)P_t \tilde{\boldsymbol{g}}_t \tilde{\boldsymbol{g}}_t^\top P_t}{(d + 1)(1 + \alpha_t)}\right); \quad (10)$$

11:     **else if** $t \leq T_0$ **then**

12:         Compute $\boldsymbol{v}_1$ and $\boldsymbol{v}_d$, the two eigenvectors associated with the largest and smallest eigenvalues of $V_t$, and pick $(\bar{\boldsymbol{a}}_{0,t}, \bar{\boldsymbol{a}}_{1,t}) = D_0(\frac{4}{5}\boldsymbol{v}_1 \pm \frac{3}{5}\boldsymbol{v}_d)$;

13:         Recommend $(\boldsymbol{a}_{0,t}, \boldsymbol{a}_{1,t})$, observe user's choice $\boldsymbol{a}_t$;

14:         $(\boldsymbol{x}_{t+1}, P_{t+1}) = (\boldsymbol{x}_t, P_t)$;

15:     **else**

16:         Compute $\boldsymbol{a}_t = \arg\max_{\boldsymbol{a} \in \mathcal{A}} \boldsymbol{u}_1^{(t)\top} \boldsymbol{a}$;

17:         Recommend $(\boldsymbol{a}_t, \boldsymbol{a}_t)$;

18:         $(\boldsymbol{x}_{t+1}, P_{t+1}) = (\boldsymbol{x}_t, P_t)$;

19:     **end if**

20:     Update $V_{t+1} = V_t + \boldsymbol{a}_t \boldsymbol{a}_t^\top$.

21: **end while**

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

# Appendix to "Learning from a Learning User for Optimal Recommendations"

## A. Preliminaries on Ellipsoid Method

A $d \times d$ matrix $A$ is symmetric when $A = A^\top$, and any symmetric matrix $A$ admits an eigenvalue decomposition $A = U\Sigma U^\top$, where $U$ is a orthogonal matrix and $\Sigma = \text{diag}(\sigma_1, \cdots, \sigma_d)$ is a diagonal matrix with diagonal elements $\sigma_1 \geq \cdots \geq \sigma_d$. We refer to $\sigma_i(A)$ as the $i$-th largest eigenvalue of $A$. A symmetric matrix $A$ is called positive definite (PD) if all its eigenvalues are strictly positive.

$$\{\boldsymbol{g}^\top(\boldsymbol{z} - \boldsymbol{x}) \leq b\} \cap \mathcal{E}'(\boldsymbol{x}', P')$$

An ellipsoid is a subset of $\mathbb{R}^d$ defined as

$$\mathcal{E}(\boldsymbol{x}, P) = \{\boldsymbol{z} | (\boldsymbol{z} - \boldsymbol{x})^\top P^{-1}(\boldsymbol{z} - \boldsymbol{x}) \leq 1\},$$

where $\boldsymbol{x} \in \mathbb{R}^d$ specifies its center and the PD matrix $P$ specifies its geometric shape. Each of the $d$ radii of $\mathcal{E}(\boldsymbol{x}, P)$ corresponds to the square root of an eigenvalue of $P$ and the volume of the ellipsoid is given by

$$\text{Vol}(\mathcal{E}(\boldsymbol{x}, P)) = V_d\sqrt{\det P} = V_d\sqrt{\prod_{i=1}^{d}\sigma_d},$$

where $V_d$ is a constant that represents the volume of the unit ball in $\mathbb{R}^d$. If a hyperplane $\boldsymbol{g}^\top(\boldsymbol{z} - \boldsymbol{x}) = b$ with normal direction $\boldsymbol{g}$ and intersection $b$ cuts the ellipsoid $\mathcal{E}(\boldsymbol{x}, P)$ to two pieces, the smallest ellipsoid containing the area $\{\boldsymbol{g}^\top(\boldsymbol{z} - \boldsymbol{x}) \leq b\} \cap \mathcal{E}(\boldsymbol{x}, P)$ can be captured by $\mathcal{E}'(\boldsymbol{x}', P')$, where the new center $\boldsymbol{x}'$ and the shape matrix $P'$ can be computed via the following closed form formula:

$$\boldsymbol{x}' = \boldsymbol{x} - \frac{1 + d\alpha}{d + 1}P\tilde{\boldsymbol{g}}, \tag{11}$$

$$P' = \frac{d^2(1 - \alpha^2)}{d^2 - 1}\left(P - \frac{2(1 + d\alpha)}{(d + 1)(1 + \alpha)}P\tilde{\boldsymbol{g}}\tilde{\boldsymbol{g}}^\top P\right), \tag{12}$$

$$\alpha = -\frac{b}{\sqrt{\boldsymbol{g}^\top P\boldsymbol{g}}}, \tag{13}$$

$$\tilde{\boldsymbol{g}} = \frac{1}{\sqrt{\boldsymbol{g}^\top P\boldsymbol{g}}}\boldsymbol{g}, \tag{14}$$

where $\alpha$ represents the cutting-depth which we will elaborate on later. To narrow down the feasible region of the target parameters, it is desirable to let $\text{Vol}(\mathcal{E}')$ as small as possible. At least, we need to ensure that $\text{Vol}(\mathcal{E}') < \text{Vol}(\mathcal{E})$. Basic algebraic calculation shows that

$$\frac{\text{Vol}(\mathcal{E}')}{\text{Vol}(\mathcal{E})} = \sqrt{\frac{\det P'}{\det P}} = \left(\frac{d^2(1 - \alpha^2)}{d^2 - 1}\right)^{\frac{d}{2}}\left(1 - \frac{2(1 + d\alpha)}{(d + 1)(1 + \alpha)}\right)^{\frac{1}{2}} \tag{15}$$

$$= \left(1 + \frac{1 + d\alpha}{d - 1}\right)^{\frac{d-1}{2}}\left(1 - \frac{1 + d\alpha}{d + 1}\right)^{\frac{d+1}{2}}$$

$$= \left(\frac{d(1 + \alpha)}{d - 1}\right)^{\frac{d-1}{2}}\left(\frac{d(1 - \alpha)}{d + 1}\right)^{\frac{d+1}{2}}, \tag{16}$$

where Eq (15) is from Eq (12) and the fact that $\det(P - \beta\boldsymbol{v}\boldsymbol{v}^\top) = (1 - \beta\|\boldsymbol{v}\|_P^2)\det(P)$. Eq (16) indicates that $\text{Vol}(\mathcal{E}') < \text{Vol}(\mathcal{E})$ if and only if $\alpha \in (-\frac{1}{d}, 1)$. The quantity $\alpha$ serves as an indicator of the "depth" of the cut: $\alpha \in (-\frac{1}{d}, 0)$ corresponds to a shallow-cut where the proposed cutting hyperplane removes less than half of the volume of the ellipsoid; $\alpha \in (0, 1)$ corresponds to a deep-cut where more than half of the volume is removed. And $\alpha = 0$ happens only when $b = 0$, meaning the cutting hyperplane goes through the center $\boldsymbol{x}$ and exactly half of the volume is removed. In our problem setting, since

we need to deal with the uncertainty in the user's response, we may only expect shallow-cuts. In addition, from Eq (16) we can show that for any $-\frac{1}{d} < \alpha < 1$,

$$\frac{\text{Vol}(\mathcal{E}')}{\text{Vol}(\mathcal{E})} \le \exp\Big( - \frac{(1+d\alpha)^2}{2d} \Big). \tag{17}$$

## B. Omitted Proofs in Section 4.1

To prove Theorem 4.1, we need the following technical lemmas. Lemma B.1 states that the product of the largest two eigenvalues of $P_t$ must shrink w.r.t. a constant factor after each cut. Since $\det(P_t)$ approaches zero at an exponential rate (from Eq (17)), $P_t$ can only have one potentially large eigenvalue while all other eigenvalues must approach zero. Lemma B.2 implies that at any time step $t$, the "gap" between $P_t$'s second-largest eigenvalue and the smallest eigenvalue can be upper bounded by a constant. Given that the determinant of $P_t$ converges to 0 at an exponential rate, all the eigenvalues of $P_t$ except the largest one must also converge to 0 exponentially fast.

**Lemma B.1.** *In Algorithm 1, let the eigenvalues of $P_t$ be $\sigma_1 \ge \cdots \ge \sigma_d$ and the eigenvalues of $P_{t+1}$ be $\{\sigma_1', \cdots, \sigma_d'\}$. Then we have*

1. *for any $3 \le i \le d$, we have equalities*

$$\sigma_i' = \frac{d^2}{d^2 - 1}\sigma_i.$$

2. *for $\sigma_1', \sigma_2'$, we have $\frac{\sigma_1'\sigma_2'}{\sigma_1\sigma_2} = \frac{d^4}{(d+1)^3(d-1)} < 1$ and the following bound*

$$\max\{\sigma_1', \sigma_2'\} \in [\frac{d^2}{(d+1)^2}\sigma_1, \frac{d^2}{d^2-1}\sigma_1], \tag{18}$$

$$\min\{\sigma_1', \sigma_2'\} \in [\frac{d^2}{(d+1)^2}\sigma_2, \frac{d^2}{d^2-1}\sigma_2]. \tag{19}$$

*Proof.* **Claim 1.** Suppose $P_t = U\Sigma U^\top$, where $\Sigma = \text{diag}(\sigma_1, \cdots, \sigma_d)$ and $U = [\boldsymbol{u}_1, \cdots, \boldsymbol{u}_d]$. From the update rule of $P_{t+1}$, for any $3 \le i \le d$ we have

$$P_{t+1}\boldsymbol{u}_i = \frac{d^2}{d^2-1}\Big(P_t - \frac{2}{d+1}P_t\tilde{\boldsymbol{g}}_t\tilde{\boldsymbol{g}}_t^\top P_t\Big)\boldsymbol{u}_i$$

$$= \frac{d^2}{d^2-1}\sigma_i\boldsymbol{u}_i - \frac{d^2}{d^2-1}\cdot\frac{2\sigma_i}{d+1}P_t\tilde{\boldsymbol{g}}_t(\tilde{\boldsymbol{g}}_t^\top\boldsymbol{u}_i)$$

$$= \frac{d^2}{d^2-1}\sigma_i\boldsymbol{u}_i, \tag{20}$$

where Eq (20) holds because $\tilde{\boldsymbol{g}}_t \in \text{span}\{\boldsymbol{u}_1, \boldsymbol{u}_2\}$. Therefore, $\{\frac{d^2}{d^2-1}\sigma_i\}_{i=3}^d$ are $d-2$ eigenvalues of $P_{t+1}$.

**Claim 2.** By the choice of $\boldsymbol{g}_t$, the cutting hyper plane always goes through $\boldsymbol{x}_t$ (i.e., $\alpha = 0$). Therefore, by Eq (17) we obtain $\frac{\prod_{i=1}^d \sigma_i'}{\prod_{i=1}^d \sigma_i} = \frac{d^2}{(d+1)^2}\cdot\Big(\frac{d^2}{d^2-1}\Big)^{d-1}$. Consider Eq (20), we conclude that the remaining two eigenvalues of $P_{t+1}$ satisfy

$$\frac{\sigma_1'\sigma_2'}{\sigma_1\sigma_2} = \frac{d^2}{(d+1)^2}\cdot\frac{d^2}{d^2-1} = \frac{d^4}{(d+1)^3(d-1)} < 1. \tag{21}$$

Next we derive the bound for $\sigma_1', \sigma_2'$. Let $\boldsymbol{g}_t = p\boldsymbol{u}_1 + q\boldsymbol{u}_2$, and

$$P_t\tilde{\boldsymbol{g}}_t = \frac{p\sigma_1}{\sqrt{p^2\sigma_1 + q^2\sigma_2}}\boldsymbol{u}_1 + \frac{q\sigma_2}{\sqrt{p^2\sigma_1 + q^2\sigma_2}}\boldsymbol{u}_2 \triangleq v_1\boldsymbol{u}_1 + v_2\boldsymbol{u}_2.$$

It is easy to see that $\frac{d^2-1}{d^2}\sigma_1', \frac{d^2-1}{d^2}\sigma_2'$ are the two eigenvalues of the following $2 \times 2$ matrix

$$A = \begin{bmatrix} \sigma_1 & 0 \\ 0 & \sigma_2 \end{bmatrix} - \frac{2}{d+1}\begin{bmatrix} v_1 \\ v_2 \end{bmatrix}\cdot\begin{bmatrix} v_1 & v_2 \end{bmatrix}. \tag{22}$$

Without loss of generality, we assume $\sigma_1' \geq \sigma_2'$. Applying Weyl's inequality in matrix theory (Fan, 1949; Bunch et al., 1978) to matrix $A$ yields

$$\sigma_1 \geq \frac{d^2-1}{d^2}\sigma_1' \geq \sigma_2 \geq \frac{d^2-1}{d^2}\sigma_2'. \tag{23}$$

On the other hand, from Eq (21) we also have

$$\frac{\sigma_1'}{\sigma_1} = \frac{d^4}{(d+1)^3(d-1)}\frac{\sigma_2}{\sigma_2'} \geq \frac{d^4}{(d+1)^3(d-1)} \cdot \frac{d^2-1}{d^2} = \frac{d^2}{(d+1)^2}, \tag{24}$$

$$\frac{\sigma_2'}{\sigma_2} = \frac{d^4}{(d+1)^3(d-1)}\frac{\sigma_1}{\sigma_1'} \geq \frac{d^4}{(d+1)^3(d-1)} \cdot \frac{d^2-1}{d^2} = \frac{d^2}{(d+1)^2}. \tag{25}$$

From Eq (23), (24), (25), we obtain Eq (18), (19) and therefore complete the proof. □

**Lemma B.2.** *At each time step $t$ in Algorithm 1, let the eigenvalue of $P_t$ be $\sigma_1^{(t)} \geq \cdots \geq \sigma_d^{(t)}$. Further let $D_t = \sigma_2^{(t)}/\sigma_d^{(t)}$, we claim*

1. *for any $t \geq 0$, $D_{t+1} \leq \frac{d+1}{d-1} \cdot D_t$;*

2. *if $D_t > \frac{d+1}{d-1}$, $D_{t+1} \leq D_t$.*

3. *for any $n \geq 0$,*

$$\max_{0 \leq t \leq n} D_t \leq \left(\frac{d+1}{d-1}\right)^2. \tag{26}$$

*Proof.* From Lemma B.1, we know that the eigenvalues of $P_{t+1}$ is $\{\sigma_1', \sigma_2', \frac{d^2}{d^2-1}\sigma_3^{(t)}, \cdots, \frac{d^2}{d^2-1}\sigma_d^{(t)}\}$, where $\sigma_1' \geq \sigma_2'$ and

$$\frac{d^2}{(d+1)^2}\sigma_2^{(t)} \leq \sigma_2' \leq \frac{d^2}{d^2-1}\sigma_2^{(t)} \tag{27}$$

**Claim 1.** Because $\sigma_1' \geq \sigma_2'$, $\sigma_3^{(t)} \geq \cdots \geq \sigma_d^{(t)}$, and note that $\sigma_2^{(t+1)}$ and $\sigma_d^{(t+1)}$ are the second-largest element and the smallest element of $\{\sigma_1', \sigma_2', \frac{d^2}{d^2-1}\sigma_3^{(t)}, \cdots, \frac{d^2}{d^2-1}\sigma_d^{(t)}\}$, the value of $(\sigma_2^{(t+1)}, \sigma_d^{(t+1)})$ must satisfy one of the following situation:

1. if $(\sigma_2^{(t+1)}, \sigma_d^{(t+1)}) = (\sigma_2', \frac{d^2}{d^2-1}\sigma_d^{(t)})$, from Eq (27) we have

$$\frac{D_{t+1}}{D_t} = \frac{d^2-1}{d^2} \cdot \frac{\sigma_2'}{\sigma_2} \leq 1. \tag{28}$$

2. if $(\sigma_2^{(t+1)}, \sigma_d^{(t+1)}) = (\frac{d^2}{d^2-1}\sigma_i^{(t)}, \frac{d^2}{d^2-1}\sigma_d^{(t)})$ for some $3 \leq i \leq d-1$, we have

$$\frac{D_{t+1}}{D_t} = \frac{\sigma_i^{(t)}/\sigma_d^{(t)}}{\sigma_2^{(t)}/\sigma_d^{(t)}} \leq 1. \tag{29}$$

3. if $(\sigma_2^{(t+1)}, \sigma_d^{(t+1)}) = (\frac{d^2}{d^2-1}\sigma_i^{(t)}, \sigma_2')$ for some $3 \leq i \leq d-1$, from Eq (27) we have

$$\frac{D_{t+1}}{D_t} = \frac{d^2}{d^2-1} \cdot \frac{\sigma_i^{(t)}}{\sigma_2'} \cdot \frac{\sigma_d^{(t)}}{\sigma_2^{(t)}} \leq \frac{d^2}{d^2-1} \cdot \frac{\sigma_2^{(t)}}{\sigma_2'} \leq \frac{d^2}{d^2-1} \cdot \frac{(d+1)^2}{d^2} = \frac{d+1}{d-1}. \tag{30}$$

By Eq (28), (29), (30), the first claim holds.

**Claim 2.** It suffices to show that the situation (3) cannot happen when $D_t > \frac{d+1}{d-1}$. In fact, when $D_t > \frac{d+1}{d-1}$, from Eq (27) we have

$$\sigma_2' \geq \frac{d^2}{(d+1)^2}\sigma_2^{(t)} = \frac{d^2}{(d+1)^2}\sigma_d^{(t)}D_t > \frac{d^2}{(d+1)^2} \cdot \frac{d+1}{d-1} \cdot \sigma_d^{(t)} = \frac{d^2}{d^2-1}\sigma_d^{(t)},$$

meaning $\sigma_2'$ cannot be the smallest eigenvalue of $P_{t+1}$. As a result, the second claim holds by Eq (28), (29).

**Claim 3.** We prove Eq (26) by contradiction. Let $n_0$ be the smallest index in set $\arg\max_{0\leq t\leq n} D_t$. If $n_0 = 0$, we have $\max_{0\leq t\leq n} D_t = D_0 = 1 < \left(\frac{d+1}{d-1}\right)^2$. Now consider the case $n_0 \geq 1$ and suppose $D_{n_0} > \left(\frac{d+1}{d-1}\right)^2$. By Claim 1, we have $D_{n_0-1} \geq \frac{d-1}{d+1} D_{n_0} > \frac{d+1}{d-1}$. Apply Claim 2 to $D_{n_0-1}$, we obtain $D_{n_0} \leq D_{n_0-1}$, which contradicts the definition of $n_0$. Hence, Claim 3 holds. $\qquad\square$

Now we are ready to present the proof of the convergence theorem for Algorithm 1:

**Theorem B.3.** *At each time step $t$ in Algorithm 1, let the eigenvalues of $P_t$ be $\sigma_1^{(t)} \geq \cdots \geq \sigma_d^{(t)}$. For any $d > 1, T > 0$, we have*

    *1. for any $2 \leq i \leq d$,*

$$\sigma_i^{(T)} \leq \exp\left(\frac{4}{d} - \frac{T}{d^2}\right). \tag{31}$$

    *2. the $\ell_2$ estimation error for $\theta_*$ is given by*

$$\left\|\theta_* - \hat{\theta}_T\right\|_2 \leq 2\sqrt{d-1}\exp\left(\frac{2}{d} - \frac{T}{2d^2}\right), \tag{32}$$

*Proof.* Since the depth of the cut $\alpha = 0$ through out the execution of Algorithm 1, from Eq (17) we have

$$\prod_{i=1}^{d} \sigma_i^{(T)} = \frac{\det P_n}{\det P_0} \leq \exp\left(-\frac{T}{d}\right). \tag{33}$$

From Lemma B.2, we have $\sigma_i^{(T)} \geq \sigma_d^{(T)} \geq \left(\frac{d-1}{d+1}\right)^2 \cdot \sigma_2^{(n)}, \forall 3 \leq i \leq d$. Therefore,

$$
\begin{aligned}
\exp\left(-\frac{T}{d}\right) &\geq \prod_{i=1}^{d} \sigma_i^{(T)} \\
&\geq \sigma_2^{(T)} \cdot \sigma_2^{(T)} \cdot \left[\left(\frac{d-1}{d+1}\right)^2 \cdot \sigma_2^{(T)}\right]^{d-2} \\
&= [\sigma_2^{(T)}]^d \cdot \left(1 - \frac{2}{d+1}\right)^{2d-4} \\
&\geq \exp(-4) \cdot [\sigma_2^{(T)}]^d.
\end{aligned}
$$

Rearranging terms yields $\sigma_2^{(T)} \leq \exp\left(\frac{4}{d} - \frac{T}{d^2}\right)$, and thus $\sigma_i^{(T)} \leq \exp\left(\frac{4}{d} - \frac{T}{d^2}\right), \forall 2 \leq i \leq d$.

Let $\langle x, y \rangle = \arccos\left(\frac{x \cdot y}{\|x\| \cdot \|y\|}\right)$ denote the included angle between vector $x$ and $y$, now we are prepared to upper bound the directional estimation error $\sin\langle\hat{\theta}_T, \theta_*\rangle$. First of all, note that $\theta_*, \mathbf{0} \in \mathcal{E}_T$ for any $n \geq 0$, meaning there exists $\{(p_i, q_i)\}_{i=1}^{d}$ such that

$$\theta_* = x_T + \sum_{i=1}^{d} p_i u_i^{(T)}, \sum_{i=1}^{d} \frac{p_i^2}{\sigma_i^{(T)}} \leq 1. \tag{34}$$

$$\mathbf{0} = x_T + \sum_{i=1}^{d} q_i u_i^{(T)}, \sum_{i=1}^{d} \frac{q_i^2}{\sigma_i^{(T)}} \leq 1. \tag{35}$$

As a result, $\theta_* = \sum_{i=1}^{d}(p_i - q_i)u_i^{(T)}$, and $p_i, q_i \leq \sqrt{\sigma_i^{(T)}}, 2 \leq i \leq d$. Therefore,

$$\sin\langle\theta_*,\hat{\theta}_T\rangle = \sqrt{1-\cos^2\langle\theta_*,\hat{\theta}_T\rangle} = \sqrt{1-\frac{(\theta_*^\top \boldsymbol{u}_1)^2}{\|\theta_*\|_2^2}} = \frac{1}{\|\theta_*\|_2}\cdot\sqrt{\sum_{i=2}^d (p_i-q_i)^2}$$

$$\le \frac{2}{\|\theta_*\|_2}\cdot\sqrt{\sum_{i=2}^d \sigma_i^{(T)}},$$

Now we know that the directional inference error for $\theta_*$ converges to zero at rate $O\big(d^{\frac{1}{2}}\exp\big(-\frac{T}{2d^2}\big)\big)$. When the system knows $\|\theta_*\|_2 = 1$, the $\ell_2$ estimation error for $\theta_*$ can be obtained from

$$\left\|\theta_* - \|\theta_*\|_2\cdot\frac{\hat{\theta}_T}{\|\hat{\theta}_T\|_2}\right\|_2 \le 2\|\hat{\theta}_T\|_2\sin(\langle\theta_*,\hat{\theta}_T\rangle/2)$$

$$\le 2\sqrt{\sum_{i=2}^d \sigma_i^{(T)}} \tag{36}$$

where the last inequality holds because $\sin x \le x, \forall x > 0$. In particular, plugin Eq (4) into the R.H.S. of Eq (36), we obtain Eq (5).

$\square$

## C. Omitted Proofs in Section 4.2

The following Lemma C.1 and C.2 are used in the proof of Theorem 4.5. Lemma C.1 and C.2 are generalizations of Lemma B.1 and B.2 under arbitrary cutting depth $\alpha_t$.

**Lemma C.1.** *In Algorithm 2, suppose a valid cut is executed at step $t$ with depth $-\frac{1}{kd} \le \alpha_t \le 0$. Let the eigenvalues of $P_t$ be $\sigma_1 \ge \cdots \ge \sigma_d$ and the eigenvalues of $P_{t+1}$ be $\{\sigma_1', \cdots, \sigma_d'\}$. Then we have*

1. *for any $3 \le i \le d$, we have equalities*

$$\sigma_i' = \frac{d^2(1-\alpha_t^2)}{d^2-1}\sigma_i.$$

2. *for $\sigma_1', \sigma_2'$, we have $\frac{\sigma_1'\sigma_2'}{\sigma_1\sigma_2} = \frac{d^4(1-\alpha_t)^3(1+\alpha_t)}{(d+1)^3(d-1)} < 1$ and the following bound*

$$\max\{\sigma_1', \sigma_2'\} \in \left[\frac{d^2(1-\alpha_t)^2}{(d+1)^2}\sigma_1, \frac{d^2(1-\alpha_t^2)}{d^2-1}\sigma_1\right], \tag{37}$$

$$\min\{\sigma_1', \sigma_2'\} \in \left[\frac{d^2(1-\alpha_t)^2}{(d+1)^2}\sigma_2, \frac{d^2(1-\alpha_t^2)}{d^2-1}\sigma_2\right]. \tag{38}$$

*Proof.* **Claim 1.** Suppose $P_t = U\Sigma U^\top$, where $\Sigma = \text{diag}(\sigma_1, \cdots, \sigma_d)$ and $U = [\boldsymbol{u}_1, \cdots, \boldsymbol{u}_d]$. From the update rule of $P_{t+1}$, for any $3 \le i \le d$ we have

$$\begin{aligned}
P_{t+1}\boldsymbol{u}_i &= \frac{d^2(1-\alpha_t^2)}{d^2-1}\left(P_t - \frac{2(1+d\alpha_t)}{(d+1)(1+\alpha_t)}P_t\tilde{\boldsymbol{g}}_t\tilde{\boldsymbol{g}}_t^\top P_t\right)\boldsymbol{u}_i \\
&= \frac{d^2(1-\alpha_t^2)}{d^2-1}\sigma_i\boldsymbol{u}_i - \frac{d^2(1-\alpha_t^2)}{d^2-1}\cdot\frac{2(1+d\alpha_t)\sigma_i}{(d+1)(1+\alpha_t)}P_t\tilde{\boldsymbol{g}}_t(\tilde{\boldsymbol{g}}_t^\top\boldsymbol{u}_i) \\
&= \frac{d^2(1-\alpha_t^2)}{d^2-1}\sigma_i\boldsymbol{u}_i, \tag{39}
\end{aligned}$$

where Eq (39) holds because $\tilde{\boldsymbol{g}}_t \in \text{span}\{\boldsymbol{u}_1, \boldsymbol{u}_2\}$. Therefore, $\{\frac{d^2(1-\alpha_t^2)}{d^2-1}\sigma_i\}_{i=3}^d$ constitute $d-2$ eigenvalues of $P_{t+1}$.

**Claim 2.** From Eq (17) we have $\frac{\prod_{i=1}^{d} \sigma_i'}{\prod_{i=1}^{d} \sigma_i} = \frac{d^2(1-\alpha_t)^2}{(d+1)^2} \cdot \left( \frac{d^2(1-\alpha_t^2)}{d^2-1} \right)^{d-1}$. Consider Eq (39), we conclude that the remaining two eigenvalues of $P_{t+1}$ satisfy

$$\frac{\sigma_1' \sigma_2'}{\sigma_1 \sigma_2} = \frac{d^2(1-\alpha_t)^2}{(d+1)^2} \cdot \frac{d^2(1-\alpha_t^2)}{d^2-1} = \frac{d^4(1-\alpha_t)^3(1+\alpha_t)}{(d+1)^3(d-1)} < 1. \tag{40}$$

Next we derive the bound for $\sigma_1', \sigma_2'$. Let $\boldsymbol{g}_t = p\boldsymbol{u}_1 + q\boldsymbol{u}_2$, and

$$P_t \tilde{\boldsymbol{g}}_t = \frac{p\sigma_1}{\sqrt{p^2 \sigma_1 + q^2 \sigma_2}} \boldsymbol{u}_1 + \frac{q\sigma_2}{\sqrt{p^2 \sigma_1 + q^2 \sigma_2}} \boldsymbol{u}_2 \triangleq v_1 \boldsymbol{u}_1 + v_2 \boldsymbol{u}_2.$$

It is easy to see that $\frac{d^2-1}{d^2(1-\alpha_t^2)}\sigma_1'$, $\frac{d^2-1}{d^2(1-\alpha_t^2)}\sigma_2'$ are the two eigenvalues of the following $2 \times 2$ matrix

$$A = \begin{bmatrix} \sigma_1 & 0 \\ 0 & \sigma_2 \end{bmatrix} - \frac{2(1+d\alpha_t)}{(d+1)(1+\alpha_t)} \begin{bmatrix} v_1 \\ v_2 \end{bmatrix} \cdot \begin{bmatrix} v_1 & v_2 \end{bmatrix}. \tag{41}$$

Without loss of generality, we assume $\sigma_1' \geq \sigma_2'$. Applying Weyl's inequality in matrix theory (Fan, 1949; Bunch et al., 1978) to matrix $A$ yields

$$\sigma_1 \geq \frac{d^2-1}{d^2(1-\alpha_t^2)}\sigma_1' \geq \sigma_2 \geq \frac{d^2-1}{d^2(1-\alpha_t^2)}\sigma_2'. \tag{42}$$

On the other hand, from Eq (40) we also have

$$\frac{\sigma_1'}{\sigma_1} = \frac{d^4(1-\alpha_t)^3(1+\alpha_t)}{(d+1)^3(d-1)} \frac{\sigma_2}{\sigma_2'} \geq \frac{d^4(1-\alpha_t)^3(1+\alpha_t)}{(d+1)^3(d-1)} \cdot \frac{d^2-1}{d^2(1-\alpha_t^2)} = \frac{d^2(1-\alpha_t)^2}{(d+1)^2}, \tag{43}$$

$$\frac{\sigma_2'}{\sigma_2} = \frac{d^4(1-\alpha_t)^3(1+\alpha_t)}{(d+1)^3(d-1)} \frac{\sigma_1}{\sigma_1'} \geq \frac{d^4(1-\alpha_t)^3(1+\alpha_t)}{(d+1)^3(d-1)} \cdot \frac{d^2-1}{d^2(1-\alpha_t^2)} = \frac{d^2(1-\alpha_t)^2}{(d+1)^2}. \tag{44}$$

From Eq (42), (43), (44), we obtain Eq (18), (19) and therefore complete the proof. $\square$

Lemma C.1 characterizes the convergence of $P_t$: the product of the largest two eigenvalues shrinks by a constant factor after each step. Since $\det(P_t)$ approaches zero at an exponential rate (from Eq (17)), $P_t$ can only have one potentially large eigenvalue while all other eigenvalues must approach zero. We formalize the claim in the following Lemma C.2.

**Lemma C.2.** *Suppose a valid cut is executed at step $t$ with depth $-\frac{1}{kd} \leq \alpha_t \leq 0$ in Algorithm 2. Let the eigenvalue of $P_t$ be $\sigma_1^{(t)} \geq \cdots \geq \sigma_d^{(t)}$. Further let $D_t = \sigma_2^{(t)}/\sigma_d^{(t)}$, we claim*

*1. for any $t \geq 0$, $D_{t+1} \leq \frac{(d+1)(1+\alpha_t)}{(d-1)(1-\alpha_t)} \cdot D_t$;*

*2. if $D_t > \frac{(d+1)(1+\alpha_t)}{(d-1)(1-\alpha_t)}$, $D_{t+1} \leq D_t$.*

*3. for any $n \geq 0$,*

$$\max_{0 \leq t \leq n} D_t \leq \left( \frac{d+1}{d-1} \right)^2. \tag{45}$$

*Proof.* From Lemma C.1, we know that the eigenvalues of $P_{t+1}$ is $\{\sigma_1', \sigma_2', \frac{d^2(1-\alpha_t^2)}{d^2-1}\sigma_3^{(t)}, \cdots, \frac{d^2(1-\alpha_t^2)}{d^2-1}\sigma_d^{(t)}\}$, where $\sigma_1' \geq \sigma_2'$ and

$$\frac{d^2(1-\alpha_t)^2}{(d+1)^2}\sigma_2^{(t)} \leq \sigma_2' \leq \frac{d^2(1-\alpha_t^2)}{d^2-1}\sigma_2^{(t)} \tag{46}$$

**Claim 1.** Because $\sigma_1' \geq \sigma_2'$, $\sigma_3^{(t)} \geq \cdots \geq \sigma_d^{(t)}$, and note that $\sigma_2^{(t+1)}$ and $\sigma_d^{(t+1)}$ are the second-largest element and the smallest element of $\{\sigma_1', \sigma_2', \frac{d^2(1-\alpha_t^2)}{d^2-1}\sigma_3^{(t)}, \cdots, \frac{d^2(1-\alpha_t^2)}{d^2-1}\sigma_d^{(t)}\}$, the value of $(\sigma_2^{(t+1)}, \sigma_d^{(t+1)})$ must satisfy one of the following situation:

1. if $(\sigma_2^{(t+1)}, \sigma_d^{(t+1)}) = (\sigma_2', \frac{d^2(1-\alpha_t^2)}{d^2-1}\sigma_d^{(t)})$, from Eq (46) we have

$$\frac{D_{t+1}}{D_t} = \frac{d^2-1}{d^2(1-\alpha_t^2)} \cdot \frac{\sigma_2'}{\sigma_2^{(t)}} \leq 1. \tag{47}$$

2. if $(\sigma_2^{(t+1)}, \sigma_d^{(t+1)}) = (\frac{d^2(1-\alpha_t^2)}{d^2-1}\sigma_i^{(t)}, \frac{d^2(1-\alpha_t^2)}{d^2-1}\sigma_d^{(t)})$ for some $3 \leq i \leq d-1$, we have

$$\frac{D_{t+1}}{D_t} = \frac{\sigma_i^{(t)}/\sigma_d^{(t)}}{\sigma_2^{(t)}/\sigma_d^{(t)}} \leq 1. \tag{48}$$

3. if $(\sigma_2^{(t+1)}, \sigma_d^{(t+1)}) = (\frac{d^2(1-\alpha_t^2)}{d^2-1}\sigma_i^{(t)}, \sigma_2')$ for some $3 \leq i \leq d-1$, from Eq (46) we have

$$\frac{D_{t+1}}{D_t} = \frac{d^2(1-\alpha_t^2)}{d^2-1} \cdot \frac{\sigma_i^{(t)}}{\sigma_2'} \cdot \frac{\sigma_d^{(t)}}{\sigma_2^{(t)}} \leq \frac{d^2(1-\alpha_t^2)}{d^2-1} \cdot \frac{\sigma_2^{(t)}}{\sigma_2'} \leq \frac{d^2(1-\alpha_t^2)}{d^2-1} \cdot \frac{(d+1)^2}{d^2(1-\alpha_t)^2} = \frac{(d+1)(1+\alpha_t)}{(d-1)(1-\alpha_t)}. \tag{49}$$

By Eq (47), (48), (49), the first claim holds.

**Claim 2.** It suffices to show that the situation (3) cannot happen when $D_t > \frac{d+1}{d-1}$. In fact, when $D_t > \frac{d+1}{d-1}$, from Eq (46) we have

$$\sigma_2' \geq \frac{d^2(1-\alpha_t)^2}{(d+1)^2}\sigma_2^{(t)} = \frac{d^2(1-\alpha_t)^2}{(d+1)^2}\sigma_d^{(t)}D_t > \frac{d^2(1-\alpha_t)^2}{(d+1)^2} \cdot \frac{(d+1)(1+\alpha_t)}{(d-1)(1-\alpha_t)} \cdot \sigma_d^{(t)} = \frac{d^2(1-\alpha_t^2)}{d^2-1}\sigma_d^{(t)},$$

meaning $\sigma_2'$ cannot be the smallest eigenvalue of $P_{t+1}$. As a result, the second claim holds by Eq (47), (48).

**Claim 3.** We prove Eq (45) by contradiction. Let $n_0$ be the smallest index in set $\arg\max_{0 \leq t \leq n} D_t$. If $n_0 = 0$, we have $\max_{0 \leq t \leq n} D_t = D_0 = 1 < \left(\frac{d+1}{d-1}\right)^2$. Now consider the case $n_0 \geq 1$ and suppose $D_{n_0} > \left(\frac{d+1}{d-1}\right)^2$. By Claim 1 and the fact that $-\frac{1}{2d} \leq \alpha_{n_0-1} \leq 0$, we have $D_{n_0-1} \geq \frac{(d-1)(1-\alpha_{n_0-1})}{(d+1)(1+\alpha_{n_0-1})}D_{n_0} > \frac{(d+1)(1+\alpha_{n_0-1})}{(d-1)(1-\alpha_{n_0-1})}$. Apply Claim 2 to $D_{n_0-1}$, we obtain $D_{n_0} \leq D_{n_0-1}$, which contradicts the definition of $n_0$. Hence, Claim 3 holds.

$\square$

**Lemma C.3.** *With the choice of $\alpha_t$ given in Eq (6), we conclude that*

1. *After each cut step, $Vol(\mathcal{E}_{t+1}) \leq \exp\left(-\frac{(k-1)^2}{2k^2 d}\right)Vol(\mathcal{E}_t)$.*

2. *If at least $d$ exploration steps are taken during $t_0 \leq t < t_0 + n$, we have $\lambda_{\min}(V_{n+t_0}) \geq \lambda_{\min}(V_{t_0}) + \frac{4D_0}{25} - 3\epsilon_0$.*

3. *At any exploitation step $t$, the instantaneous regret is upper bounded by $2L\|\theta_* - \boldsymbol{u}_1^{(t)}\|_2^2$.*

*Proof.* **First Claim:** We first justify our choice of $\alpha_t$. With out loss of generality, assume $\boldsymbol{a}_{1,t}$ is preferred over $\boldsymbol{a}_{0,t}$, then according to the user's decision rule (3) we have

$$\theta_t^\top(\boldsymbol{a}_{0,t} - \boldsymbol{a}_{1,t}) \leq |\beta_t| \cdot (\|\boldsymbol{a}_{0,t}\|_{V_t^{-1}} + \|\boldsymbol{a}_{1,t}\|_{V_t^{-1}}) \leq c_2 t^{\gamma_2} \cdot (\|\boldsymbol{a}_{0,t}\|_{V_t^{-1}} + \|\boldsymbol{a}_{1,t}\|_{V_t^{-1}}). \tag{50}$$

Next we translate Eq (50) into the estimation with respect to $\theta_*$. According to the Estimation rule (2), with probability $1 - \delta$,

$$\begin{aligned}(\theta_* - \theta_t)^\top(\boldsymbol{a}_{0,t} - \boldsymbol{a}_{1,t}) &\leq \|\theta_* - \theta_t\|_{V_t} \cdot \|\boldsymbol{a}_{0,t} - \boldsymbol{a}_{1,t}\|_{V_t^{-1}} \\ &\leq c_1 g(\delta)t^{\gamma_1}\|\boldsymbol{a}_{0,t} - \boldsymbol{a}_{1,t}\|_{V_t^{-1}},\end{aligned}$$

and therefore according to the $(c, \gamma)-$rational assumption, we obtain

$$(\boldsymbol{a}_{0,1} - \boldsymbol{a}_{1,t})^\top(\theta_* - \boldsymbol{x}) \leq 0$$

$$\theta_*^\top (a_{0,t} - a_{1,t}) \le \theta_t^\top (a_{0,t} - a_{1,t}) + (\theta_* - \theta_t)^\top (a_{0,t} - a_{1,t})$$
$$\le c_2 t^{\gamma_2} \cdot (\|a_{0,t}\|_{V_t^{-1}} + \|a_{1,t}\|_{V_t^{-1}}) + c_1 g(\delta) t^{\gamma_1} \|a_{0,t} - a_{1,t}\|_{V_t^{-1}}$$
$$\le c t^\gamma \left( \|a_{0,t}\|_{V_t^{-1}} + \|a_{1,t}\|_{V_t^{-1}} + g(\delta) \cdot \|a_{0,t} - a_{1,t}\|_{V_t^{-1}} \right). \tag{51}$$

According to $\epsilon_0$-DC and the definition of $g_t$, we have

$$\|g_t - (a_{0,t} - a_{1,t})\|_2 \le \|a_{0,t} - \bar{a}_{0,t}\|_2 + \|a_{1,t} - \bar{a}_{1,t}\|_2 \le 2\epsilon_0. \tag{52}$$

Using Eq (52), we may relax Eq (51) by replacing $a_{0,t} - a_{1,t}$ with $g_t = \bar{a}_{0,t} - \bar{a}_{1,t}$, accounting for the error introduced by the inaccuracy of the exploration direction as below:

$$g_t^\top (\theta_* - x_t) = g_t^\top \theta_*$$
$$= (a_{0,t} - a_{1,t})^\top \theta_* + (g_t - (a_{0,t} - a_{1,t}))^\top \theta_*$$
$$\le c t^\gamma \left( \|a_{0,t}\|_{V_t^{-1}} + \|a_{1,t}\|_{V_t^{-1}} + g(\delta) \cdot \|a_{0,t} - a_{1,t}\|_{V_t^{-1}} \right) + \|g_t - a_{0,t} + a_{1,t}\|_2 \cdot \|\theta_*\|_2$$
$$\le c t^\gamma \left( \|a_{0,t}\|_{V_t^{-1}} + \|a_{1,t}\|_{V_t^{-1}} + g(\delta) \cdot \|a_{0,t} - a_{1,t}\|_{V_t^{-1}} \right) + 2\epsilon_0, \tag{53}$$

where Eq (53) holds because we assume $\|\theta_*\|_2 = 1$. Hence, by equation (13), the cutting depth

$$\alpha_t = -\frac{c t^\gamma \left( \|a_{0,t}\|_{V_t^{-1}} + \|a_{1,t}\|_{V_t^{-1}} + g(\delta) \cdot \|a_{0,t} - a_{1,t}\|_{V_t^{-1}} \right) + 2\epsilon_0}{\|g_t\|_{P_t}}. \tag{54}$$

Therefore, we may leverage Eq (54) to evaluate the cutting depth $\alpha_t$ and perform a cut whenever $\alpha_t \ge -\frac{1}{kd} > -\frac{1}{d}$ is satisfied. From Eq (17), we therefore conclude $\text{Vol}(\mathcal{E}_{t+1}) \le \exp\left( -\frac{(k-1)^2}{2k^2 d} \right) \text{Vol}(\mathcal{E}_t)$.

**Second Claim:** To prove the second claim, we need the following auxiliary lemma:

**Lemma C.4.** *A is a $d \times d$ PSD matrix with eigendecomposition $A = U diag(\sigma_1, \cdots, \sigma_d) U^T$, where $\sigma_1 \le \cdots \le \sigma_d$ and $U = [u_1, \cdots, u_d]$. For any $v \in \mathbb{R}^d$, let the eigenvalues of $A + vv^T$ be $\sigma_1' \le \cdots \le \sigma_d'$. Then we have*

1. *$\sigma_1 \le \sigma_1' \le \sigma_2 \le \sigma_2' \le \cdots \le \sigma_d \le \sigma_d' \le \sigma_d + v^T v$.*

2. *if $v = p u_1 + q u_d + \epsilon$ for some $p^2 + q^2 = 1, \|\epsilon\|_2 = \epsilon < 1$, $\{\sigma_i\}_{i=1}^d$ and $\{\sigma_i'\}_{i=1}^d$ have at least $d - 2$ common values. Furthermore, conditioned on $\sigma_d > \sigma_1 + p^2 - q^2$, at least one of the following claims is true:*
   *a) $\sigma_1' \ge \sigma_1 + p^2 - |pq| - 3\epsilon$.*
   *b) $\sigma_1' = \sigma_2$, and $\sigma_i' \ge \sigma_1 + p^2 - |pq| - 3\epsilon$ for some $2 \le i \le d$.*

*Proof.* The first claim is a direct corollary of Weyl's inequality in matrix theory (Fan, 1949; Bunch et al., 1978). Now we prove the second claim for the special case $\epsilon = 0$. From Secular Equations, we know that $\sigma_1'$ is the smallest root of the following equation

$$f(\lambda) = \prod_{i=1}^d (\sigma_i - \lambda) + p^2 \prod_{j \neq 1}^d (\sigma_j - \lambda) + q^2 \prod_{j \neq d}^d (\sigma_j - \lambda)$$
$$= \left[ (\sigma_1 - \lambda)(\sigma_d - \lambda) + p^2(\sigma_d - \lambda) + q^2(\sigma_1 - \lambda) \right] \prod_{j \neq 1, d}^d (\sigma_j - \lambda)$$
$$= \left[ \lambda^2 - (1 + \sigma_1 + \sigma_d)\lambda + q^2\sigma_1 + p^2\sigma_d + \sigma_1\sigma_d \right] \prod_{j \neq 1, d}^d (\sigma_j - \lambda).$$

Therefore, $\sigma_1'$ is the smaller one between $\sigma_2$ and the smallest root of the quadratic equation $\lambda^2 - (1 + \sigma_1 + \sigma_d) + q^2\sigma_1 + p^2\sigma_d + \sigma_1\sigma_d = 0$, i.e.,

$$\sigma_1' = \min\{\sigma_2, \frac{1 + \sigma_1 + \sigma_d - \sqrt{(1 + \sigma_1 + \sigma_d)^2 - 4(q^2\sigma_1 + p^2\sigma_d + \sigma_1\sigma_d)}}{2}\}. \tag{55}$$

Note that when $\sigma_d > \sigma_1 + p^2 - q^2$, we have

$$\frac{1 + \sigma_1 + \sigma_d - \sqrt{(1 + \sigma_1 + \sigma_d)^2 - 4(q^2\sigma_1 + p^2\sigma_d + \sigma_1\sigma_d)}}{2}$$

$$= \frac{1 + \sigma_1 + \sigma_d - \sqrt{(p^2 - q^2 + \sigma_1 - \sigma_d)^2 + 4p^2q^2}}{2}$$

$$\geq \frac{1}{2}(1 + \sigma_1 + \sigma_d - |p^2 - q^2 + \sigma_1 - \sigma_d| - 2|pq|) \tag{56}$$

$$= \sigma_1 + p^2 - |pq|, \tag{57}$$

where Eq (56) holds because $\sqrt{a^2 + b^2} \leq |a| + |b|$. From Eq (55) and Eq (57) we conclude the proof.

Next it remains to show that with a small perturbation $\epsilon$ on $v$, the change of the smallest eigenvalue will only deviate at most $3\epsilon$. From Weyl's eigenvalue perturbation inequality, for any Hermitian matrices $M, \Delta$, we have $|\lambda_k(M + \Delta) - \lambda_k(M)| \leq \|\Delta\|_2$, where $\lambda_k(\cdot)$ denotes the $k-$th largest eigenvalue of a given matrix. Using this tool, we can upper bound the difference between the smallest eigenvalues of matrix $A + vv^\top$ and $A + (v + \epsilon)(v + \epsilon)^\top$ as below:

$$\lambda_1(A + (v + \epsilon)(v + \epsilon)^\top) - \lambda_1(A + vv^\top)$$

$$\leq \|\epsilon v^\top + v\epsilon^\top + \epsilon\epsilon^\top\|_2 \leq \|\epsilon v^\top + v\epsilon^\top\|_2 + \|\epsilon\epsilon^\top\|_2$$

$$\leq 2\epsilon + \epsilon^2 < 3\epsilon, \tag{58}$$

where Eq (58) holds because for any $\|x\|_2 = 1$, $x^\top(\epsilon v^\top + v\epsilon^\top)x \leq 2\|\epsilon\|_2$ and $x^\top(\epsilon\epsilon^\top)x \leq \|\epsilon\|_2^2$. $\quad\square$

Now we are ready to prove the second claim. Without loss of generality, we consider the case $D_0 = 1$. Suppose Algorithm 2 had executed $d$ exploration steps from $t = t_0$ to $t = t_0 + n$. By the first claim of Lemma C.4, we know $\{\sigma_1^{(\tau)}\}_{\tau=1}^t$ is always non-decreasing. Therefore, it suffices to prove that after $d$ consecutive exploration steps, $\sigma_1^{(t_0+d)} \geq \sigma_1^{(t_0)} + p^2 - |pq| - 3\epsilon_0$.

From the second claim in Lemma C.4:

1. if situation $a)$ happens at least once during the $d$ exploration steps, we already obtain $\sigma_1^{(t_0+d)} \geq \sigma_1^{(t_0)} + p^2 - |pq| - 3\epsilon_0$.

2. if we always observe situation $b)$, consider the set $C_t = \{i : \sigma_i^{(t_0+t)} < \sigma_1^{(t_0)} + p^2 - |pq| - 3\epsilon_0\}$. From Lemma C.4, we can prove $|C_{t+1}| \leq |C_t| - 1$. Since $\sigma_d^{(t_0)} > \sigma_1^{(t_0)} + p^2 - |pq| - 3\epsilon_0$, we have $|C_1| \leq d - 1$. Therefore, there must exists $1 \leq k \leq d$ such that $|C_k| = 0$, meaning $\sigma_1^{(t_0+d)} \geq \sigma_1^{(k)} \geq \sigma_1^{(t_0)} + p^2 - |pq| - 3\epsilon_0$.

By taking $(p, q) = (\frac{4}{5}, \frac{3}{5})$, we obtain the desirable result.

**Thrid Claim:** Given $\|\theta_*\|_2 = 1$, denote $\hat{\theta} = u_1^{(t)}$ and $\|\theta_* - \hat{\theta}\|_2 = \epsilon$. Let $x_* = \arg\max_{x \in \mathcal{A}} x^T\theta_*$ and $\hat{x} = \arg\max_{x \in \mathcal{A}} x^T\hat{\theta}$. We have

$$\begin{aligned}
\theta_*^T(x_* - \hat{x}) &= (\theta_* - \hat{\theta})^T x_* + (x_* - \hat{x})^T\hat{\theta} + (\hat{\theta} - \theta_*)^T\hat{x} \\
&\leq (\theta_* - \hat{\theta})^T x_* + (\hat{\theta} - \theta_*)^T\hat{x} && \text{by definition of } \hat{x} \\
&= (\hat{\theta} - \theta_*)^T(\hat{x} - x_*) \\
&\leq \|\hat{\theta} - \theta_*\|_2 \cdot \|\hat{x} - x_*\|_2 && \text{by Cauchy-Schwarz} \\
&\leq L \cdot \|\hat{\theta} - \theta_*\|_2^2. && \text{by L-SRC}
\end{aligned}$$

As a result, the instantaneous regret is upper bounded by $2L\|\boldsymbol{u}_1^{(t)} - \theta_*\|_2^2$. □

Now we are ready to analyze the regret of Algorithm 2:

**Theorem C.5.** *For any $d > 1, n > 0$, let $\sigma_i^{(n)}$ be the $i$-th largest eigenvalue of $P_n$ after the $n$-th cut, we have*

1. *For any $2 \leq i \leq d$,*

$$\sigma_i^{(n)} \leq \exp\Big(\frac{4}{d} - \frac{(k-1)^2 n}{k^2 d^2}\Big). \tag{59}$$

2. *When $T_0 = O\Big(cL^{\frac{1}{2}} D_1^{\frac{1}{2}} D_0^{-\frac{3}{2}} g(\delta) d^2 T^{\frac{1}{2}+\gamma}\Big)$ and $\epsilon_0 < O\big(cD_1 D_0^{-\frac{1}{2}} d^{-\frac{1}{2}} T^{-\frac{1}{4}+\frac{\gamma}{2}}\big)$, the regret of RAES is upper bounded by $O\Big(cL^{\frac{1}{2}} D_1^{\frac{3}{2}} D_0^{-\frac{3}{2}} g(\frac{\delta}{T_0}) d^2 T^{\frac{1}{2}+\gamma}\Big)$ with probability $1 - \delta$.*

*Proof.* Since the depth of the cut $\alpha_t \geq -\frac{1}{kd}$ through out the execution of Algorithm 2, from Lemma 4.4 and Eq (17) we have

$$\prod_{i=1}^{d} \sigma_i^{(n)} = \prod_{i=0}^{n-1} \frac{\det P_{i+1}}{\det P_i} \leq \prod_{i=0}^{n-1} \frac{\text{Vol}(\mathcal{E}_{i+1})}{\text{Vol}(\mathcal{E}_i)} = \exp\Big(-\frac{(k-1)^2 n}{k^2 d}\Big). \tag{60}$$

From Lemma C.2, we have $\sigma_i^{(n)} \geq \sigma_d^{(n)} \geq \big(\frac{d-1}{d+1}\big)^2 \cdot \sigma_2^{(n)}, \forall 3 \leq i \leq d$. Therefore,

$$\exp\Big(-\frac{(k-1)^2 n}{k^2 d}\Big) \geq \prod_{i=1}^{d} \sigma_i^{(n)}$$

$$\geq \sigma_2^{(n)} \cdot \sigma_2^{(n)} \cdot \Big[\big(\frac{d-1}{d+1}\big)^2 \cdot \sigma_2^{(n)}\Big]^{d-2}$$

$$= [\sigma_2^{(n)}]^d \cdot \Big(1 - \frac{2}{d+1}\Big)^{2d-4}$$

$$\geq \exp(-4) \cdot [\sigma_2^{(n)}]^d.$$

Rearranging terms yields $\sigma_2^{(n)} \leq \exp\Big(\frac{4}{d} - \frac{(k-1)^2 n}{k^2 d^2}\Big)$, and thus $\sigma_i^{(n)} \leq \exp\Big(\frac{4}{d} - \frac{(k-1)^2 n}{k^2 d^2}\Big), \forall 2 \leq i \leq d$.

Next we show the second claim. Suppose the total number of cut during the first $T_0/2$ step is $N_0$.

1. if $N_0 \geq \frac{d^2 k^2}{(k-1)^2} \log T_0 + \frac{4dk^2}{(k-1)^2}$, from Eq (59) we have $\sigma_i^{(N_0)} \leq \frac{1}{T_0}$.

2. if $N_0 < \frac{d^2 k^2}{(k-1)^2} \log T_0 + \frac{4dk^2}{(k-1)^2}$, for sufficiently large $T$, there are at least $T_0/2 - N \geq T_0/2 - \frac{d^2 k^2}{(k-1)^2} \log T_0 - \frac{4dk^2}{(k-1)^2} > \frac{T_0}{3}$ exploration steps during the first $T_0/2$ iterations. From the second claim of Lemma 4.4, $\lambda_{\min}(V_{T_0}) \geq \frac{\beta T_0}{d}$, where $\beta = \frac{1}{3}\big(\frac{4D_0}{25} - 3\epsilon_0\big)$ is a positive constant. Using the definition of matrix norm, we have for any $t$, $\|\boldsymbol{a}_{0,t}\|_{V_t^{-1}}, \|\boldsymbol{a}_{1,t}\|_{V_t^{-1}} \leq D_1\sqrt{\lambda_{\max}(V_t^{-1})}, \|\boldsymbol{a}_{0,t} - \boldsymbol{a}_{1,t}\|_{V_t^{-1}} \leq 2D_1\sqrt{\lambda_{\max}(V_t^{-1})}$, and $\|\boldsymbol{g}_t\|_{P_t} \geq D_0(\sigma_2^{(t)})^{-\frac{1}{2}}$. Therefore, we have

$$\alpha_t \geq -2\Big[ct^\gamma D_1 D_0^{-1}\big(1 + g(\delta)\big) \cdot \sqrt{\lambda_{\max}(V_t^{-1})} + \epsilon_0 D_0^{-1}\Big] \cdot (\sigma_2^{(t)})^{-\frac{1}{2}}.$$

According to Algorithm 2, as long as we have $-2\Big[ct^\gamma D_1 D_0^{-1}\big(1 + g(\delta)\big) \cdot \sqrt{\lambda_{\max}(V_t^{-1})} + \epsilon_0 D_0^{-1}\Big] \cdot (\sigma_2^{(t)})^{-\frac{1}{2}} \geq -\frac{1}{kd}$, a cut will happen at step $t$ and we can shrink $\sqrt{\sigma_2^{(t)}}$ with probability $1 - \delta$. In other words, after the last time Algorithm 2 choose to cut during the first $T_0$ round, we have

$$\sqrt{\sigma_2^{(t)}} \leq \frac{2D_1 ckd^{1.5} t^\gamma (1 + g(\delta))}{D_0\sqrt{\beta T_0}} + \frac{2kd\epsilon_0}{D_0} < \frac{3D_1 ckd^{1.5} T_0^\gamma (1 + g(\delta))}{D_0\sqrt{\beta T_0}}, \tag{61}$$

where the last inequality holds because $\epsilon_0 < \frac{cD_1}{2\sqrt{\beta}} d^{-\frac{1}{2}} T^{-\frac{1}{4}+\frac{\gamma}{2}}$. On the other hand, the total number of cuts $n$ such that Eq (61) is satisfied is upper bounded by $O(\log T_0)$ since $\sigma_2^{(t)}$ shrinks exponentially w.r.t. the cut number $t$. Therefore, when $T$ is reasonably large, we can guarantee $n < T_0/2$ and conclude that Eq (61) holds for all $t > T_0$.

According to Eq (36) and the third claim in Lemma 4.4, when algorithm 2 enters the exploitation phase when $t > T_0$, with probability $1 - T_0\delta$, the instantaneous regret is upper bounded by

$$\theta_*^\top[(a_* - a_{0,t}) + (a_* - a_{1,t})] \leq 8(d-1) \cdot L \cdot \left(\frac{3D_1 ckd^{1.5}T_0^\gamma(1+g(\delta))}{D_0\sqrt{\beta T_0}}\right)^2 \tag{62}$$

$$\leq \frac{72D_1^2 Lc^2 k^2 d^4(1+g(\delta))^2}{\beta D_0^2 T_0^{1-2\gamma}} \tag{63}$$

For each cut or exploration step in the first $T_0$ rounds, the incurred instantaneous regret is at most $T_0 D_1$. For each following exploitation step, the regret is upper bounded by $\frac{72D_1^2 Lc^2 k^2 d^4(1+g(\delta))^2}{D_0^2 \beta T_0^{1-2\gamma}}$. Hence, we can upper bound the accumulated regret by

$$R_T \leq D_1 T_0 + \frac{72D_1^2 Lc^2 k^2 d^4(1+g(\delta))^2}{D_0^2 \beta T_0} \cdot T^{1+2\gamma}$$

$$\leq \frac{12D_1}{D_0}\sqrt{\frac{2LD_1}{\beta}} ck(1+g(\delta)) \cdot d^2 T^{\frac{1}{2}+\gamma}, \tag{64}$$

where the optimal regret is achieved when $T_0 = \frac{6ck}{D_0}\sqrt{\frac{6LD_1}{\frac{4D_0}{25}-3\epsilon_0}}(1+g(\delta))d^2 T^{\frac{1}{2}+\gamma}$, we have $R_T \leq \frac{12ckD_1}{D_0}\sqrt{\frac{6LD_1}{\frac{4D_0}{25}-3\epsilon_0}}(1+g(\delta))d^2 T^{\frac{1}{2}+\gamma}$. By applying the union bound to the first $T_0$ rounds, we thus conclude that with probability $1 - \delta$,

$$R_T \leq \frac{60D_1}{D_0}\sqrt{\frac{6LD_1}{4D_0 - 75\epsilon_0}} ck\left(1 + g(\frac{\delta}{T_0})\right) \cdot d^2 T^{\frac{1}{2}+\gamma}.$$

$\square$

## D. Omitted Proofs in Section 4.3

To derive our lower bound result, we need to leverage the minimax lower bound result for stochastic linear bandits (adapted from Theorem 24.1 in (Lattimore & Szepesvári, 2020)). For convenience, we use $\theta_{i:j}$ to denote the slice of vector $\theta$ from the $i-$th element to the $j-$th element.

**Theorem D.1.** *There exists a function $T_0(d) > 0$ such that for any $d \geq 1$, $T > T_0(d)$, and any algorithm $\mathcal{G}$ that has merely access to the comparison feedback given by a rational user defined in Definition 3.1, there exists $\theta \in \partial \mathbb{B}_1^d$ such that the expected regret $R_T$ given by Eq (1) obtained by $\mathcal{G}$ satisfies*

$$R_T^{(s)}(\mathcal{G}, \theta) \geq \frac{\exp(-2)}{4}(d-1)\sqrt{T}. \tag{65}$$

*Proof.* We prove our claim by contradiction using Theorem D.2. Essentially, we show that if the system has a powerful algorithm to achieve an expected regret lower than the RHS of Eq. (8), then we can leverage this algorithm for the linear bandit problem in Theorem D.2 with an expected regret even lower than the lower bound and thus draw the contradiction.

Suppose for any $d > 0$, there exists sufficiently large $T$ and an algorithm $\mathcal{G}$ such that for any parameter $\theta_* \in \partial \mathbb{B}_1^d$, we have

$$\mathbb{E}\left[\sum_{t=1}^T \theta_*^\top(2a_* - a_{0,t} - a_{1,t})\right] = R_T^{(s)}(\mathcal{G}, \theta_*) < \frac{\exp(-2)}{4}(d-1)\sqrt{T}.$$

As a result, the following inequalities must hold simultaneously:

$$\mathbb{E}\Big[\sum_{t=1}^{T}\theta_*^\top (a_* - a_{0,t})\Big] < \frac{\exp(-2)}{4}(d-1)\sqrt{T},$$

$$\mathbb{E}\Big[\sum_{t=1}^{T}\theta_*^\top (a_* - a_{1,t})\Big] < \frac{\exp(-2)}{4}(d-1)\sqrt{T}.$$

(66)

Now suppose a principal can observe the interaction between a user and a system equipped with algorithm $\mathcal{G}$, then he can construct two algorithms $\mathcal{G}_0, \mathcal{G}_1$ for linear bandit as follows:

---

**Algorithm $\mathcal{G}_i$ :**
**Input:** the time horizon $T$.
For $t \in [T]$:

  1. Call algorithm $\mathcal{G}$ to generate two candidates $(a_{0,t}, a_{1,t})$.
  2. Present $(a_{0,t}, a_{1,t})$ to the user and and let her decide the winner $a_{*,t}$ using decision rule 3.
  3. Return the feedback $a_{*,t}$ to algorithm $\mathcal{G}$ and update the internal state of $\mathcal{G}$ accordingly.

**Output:** the sequential decisions $\{a_{i,t}\}_{t=1}^{T}$.

---

From Eq. (66), we know that both $\mathcal{G}_0$ and $\mathcal{G}_1$ achieve an expected regret no greater than $\frac{\exp(-2)}{4}(d-1)\sqrt{T}$, which draws a contradiction to Theorem D.2.

$\square$

To prove Theorem D.1, we need the following technical lemma:

**Lemma D.2.** *Let $d \geq 2$ and $T \geq d^2$, the action set $\mathcal{A} = [-1,1]^d$ be a hypercube in $\mathbb{R}^d$, and*

$$\Theta = \Big\{\theta \in \mathbb{R}^d : \|\theta\|_1 = 1, \theta_{1:d-1} \in \{-\frac{1}{\sqrt{T}}, \frac{1}{\sqrt{T}}\}^{d-1}\Big\}.$$

*Let the expected regret for a linear bandit problem induced by any fixed algorithm $\mathcal{G}$ and parameter $\theta$ be*

$$R_T(\mathcal{G}, \theta) = T \max_{a \in \mathcal{A}} \langle a, \theta \rangle - \mathbb{E}[\sum_{t=1}^{T} \langle a_t, \theta \rangle],$$

(67)

*where the expectation is taken with respect to the randomness generated by the standard Gaussian noise $\mathcal{N}(0,1)$ in the reward. Then there must exist a parameter vector $\theta \in \Theta$ such that*

$$R_T(\mathcal{G}, \theta) \geq \frac{\exp(-2)}{8}(d-1)\sqrt{T}.$$

(68)

*Proof.* Fix an algorithm $\mathcal{G}$ and a time horizon $T$. For any $\theta \in \Theta$, let $\mathbb{P}_\theta$ be the probability measure on the probability space induced by the $T$-round interconnection of policy $\mathcal{G}$ and the problem instance given by $\theta$. Let $D(\cdot, \cdot)$ denote the relative entropy, from the general form of divergence decomposition lemma (Lemma 15.1 in (Lattimore & Szepesvári, 2020)), we have

$$D(\mathbb{P}_\theta, \mathbb{P}_{\theta'}) = \mathbb{E}_\theta \Big[ \sum_{t=1}^{T} D(\mathcal{N}(\langle a_t, \theta \rangle, 1), \mathcal{N}(\langle a_t, \theta' \rangle, 1)) \Big]$$

$$= \frac{1}{2} \sum_{t=1}^{T} \mathbb{E}_\theta[\langle a_t, \theta - \theta' \rangle^2].$$

(69)

For any $i \in [d-1]$ and $\theta \in \Theta$, let $a_{t,i}$ and $\theta_i$ be the $i$-th element of $a_t$ and $\theta$ and define

$$p_{\theta_i} = \mathbb{P}_\theta \Big( \sum_{t=1}^{T} \mathbb{I}\{\text{sign}(a_{t,i}) \neq \text{sign}(\theta_i)\} \geq \frac{T}{2} \Big).$$

Let $\theta, \theta'$ be any pair of elements in $\Theta$ such that they only differ in the $i-$th element. Therefore, by the Bretagnolle-Huber inequality (Theorem 14.2 in (Lattimore & Szepesvári, 2020)) and Eq. (69),

$$p_{\theta_i} + p_{\theta_i'} \geq \frac{1}{2} \exp \Big( - D(\mathbb{P}_\theta, \mathbb{P}_{\theta'}) \Big)$$

$$= \frac{1}{2} \exp \Big( - \frac{1}{2} \sum_{t=1}^{T} \mathbb{E}_\theta [\langle a_t, \theta - \theta' \rangle^2] \Big)$$

$$\geq \frac{1}{2} \exp \Big( - \frac{1}{2} \cdot T \Big(\frac{2}{\sqrt{T}}\Big)^2 \Big) = \frac{1}{2} \exp(-2).$$

Fix $i \in [d-1]$, there are $|\Theta| = 2^{d-1}$ such pairs $(\theta, \theta')$. Take summation over $i$ and all such pairs, we obtain

$$\sum_{\theta \in \Theta} \frac{1}{|\Theta|} \sum_{i=1}^{d} p_{\theta_i} \geq \frac{1}{|\Theta|} \sum_{i=1}^{d-1} \sum_{\theta \in \Theta} p_{\theta_i}$$

$$= \frac{1}{|\Theta|} \sum_{i=1}^{d-1} \frac{1}{2} \sum_{(\theta, \theta')} (p_{\theta_i} + p_{\theta_i'})$$

$$\geq \frac{d-1}{4} \exp(-2),$$

which implies that there exists a $\theta \in \Theta$ such that $\sum_{i=1}^{d} p_{\theta_i} \geq \frac{d-1}{4} \exp(-2)$. By the definition of $p_{\theta_i}$, the regret of $\mathcal{G}$ for this problem instance with parameter $\theta$ is at least

$$R_T(\mathcal{A}, \theta) = \mathbb{E}_\theta \Big[ \sum_{t=1}^{T} \sum_{i=1}^{d} (\text{sign}(\theta_i) - a_{t,i}) \theta_i \Big]$$

$$\geq \sqrt{\frac{1}{T}} \sum_{i=1}^{d} \mathbb{E}_\theta \Big[ \sum_{t=1}^{T} \mathbb{I}\{\text{sign}(a_{t,i}) \neq \text{sign}(\theta_i)\} \Big]$$

$$\geq \frac{\sqrt{T}}{2} \sum_{i=1}^{d} \mathbb{P}_\theta \Big( \sum_{t=1}^{T} \mathbb{I}\{\text{sign}(a_{ti}) \neq \text{sign}(\theta_i)\} \geq \frac{T}{2} \Big)$$

$$= \frac{\sqrt{T}}{2} \sum_{i=1}^{d} p_{\theta_i} \geq \frac{\exp(-2)}{8} (d-1)\sqrt{T},$$

where the first line follows since the optimal action satisfies $a_i^* = \text{sign}(\theta_i)$ and for $i \in [d]$, the first inequality follows from a simple case-based analysis showing that $(\text{sign}(\theta_i) - a_{ti})\theta_i \geq |\theta_i| \mathbb{I}\{\text{sign}(a_{ti}) \neq \text{sign}(\theta_i)\}$, the second inequality is from Markov's inequality, and the last inequality follows from the choice of $\theta$.

$\square$

# E. Additional Experiments

## E.1. Configuration of Baseline Algorithms

**Dueling Bandit Gradient Descent (DBGD)**: DBGD (Yue & Joachims, 2009) maintains the currently best candidate $a_t$ and compares it with a neighboring point $a_t + \eta u_t$ along a random direction $u_t$. An update is taken when the proposed point wins the comparison. DBGD works for continuous convex action set and has a regret guarantee of $O(T^{3/4})$. Although

its theoretical guarantee only holds under a strictly concave utility function, it can be reasonably adapted to our problem setting empirically. DBGD's hyper-parameters include the starting point $\boldsymbol{w}_0$, and two learning rates $\delta, \gamma$ that control the step-lengths for proposing new points and update the current points, respectively. In the experiment, these hyper-parameters are set to $(\boldsymbol{w}_0, \delta, \gamma) = (\boldsymbol{0}, d^{-\frac{1}{2}}T^{-\frac{1}{4}}, T^{-\frac{1}{2}})$, as recommended in (Yue & Joachims, 2009).

**Doubler**: Doubler (Ailon et al., 2014) is the first approach that converts a dueling bandit problem into a conventional multi-armed bandit (MAB) problem. Doubler proceeds in epochs of exponentially increasing size: in each epoch, the left arm is sampled from a fixed distribution, and the right arm is chosen using an MAB algorithm to minimize regret against the left arm. The feedback received by the MAB algorithm is the number of wins the right arm encounters when compared against the left arm. Doubler is proved to have $\tilde{O}(T^{1/2})$ regret for continuous action set under the linear reward assumption. The black-box MAB algorithm that is needed to initiate Doubler is set to the OFUL algorithm in (Abbasi-Yadkori et al., 2011).

**Sparring**: Sparring (Ailon et al., 2014; Sui et al., 2017) is also a general reduction from dueling bandit to MAB. Like Doubler, it also requires black-box calls to an MAB algorithm and achieves regret of the same order as the MAB algorithm. Instead of comparing with a fixed distribution, Sparring initializes two MAB instances and lets them "spar" against each other. As a heuristic improvement of Doubler, Sparring does not have a regret upper bound guarantee but is reported to enjoy a better performance compared to Doubler (Ailon et al., 2014). The black-box MAB algorithm that is needed to initiate Sparring is set to the OFUL algorithm in (Abbasi-Yadkori et al., 2011).

### E.2. Simulation Environment and Metrics

In all experiments, we fix the action set $\mathcal{A} = \mathbb{B}_2^d(0,1)$, i.e., $D_0 = D_1 = 1$, and $\delta = 0.1, k = 1.05$. We consider a $(1,\gamma)$-rational user with $\gamma \in \{0, 0.2\}$ and prior knowledge matrix $V_0$. The user's decision sequence $\{\beta_t^{(0)}\}$ and $\{\beta_t^{(1)}\}$ are independently drawn from $[-t^\gamma, t^\gamma]$. The ground-truth parameter $\theta_*$ is sampled from $\partial \mathbb{B}_2^d(0,1)$ and the reported results are collected from the same problem instance and averaged over 10 independent runs.

### E.3. Additional Results

Figure 4 shows the accumulated regret of RAES and other baselines when $d = 5$ against a $(1, 0.2)-$rational user with different $V_0$. Compared to Figure 3, we can see that RAES leads the performance compared to other baseline algorithms with a larger margin when facing a less rational user equipped a larger $\gamma$.

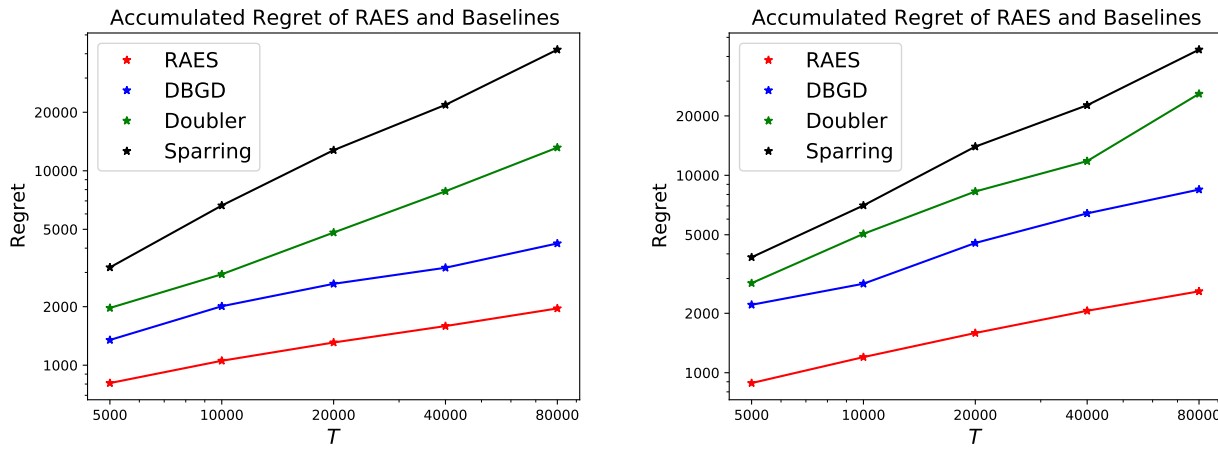

*Figure 4.* The accumulated regret of RAES and three baseline algorithms against a learning user with different $\gamma, V_0$, and $T$. Different colors specify different algorithms, and each star represents the accumulated regret (y-axis) of the algorithm given time horizon $T$ (x-axis) with $\gamma = 0.2$. Left: $V_0 = 100I_d$; right: $V_0 = \text{diag}(100, 10, 5, 2, 1)$.

Figure 5, 6 show the accumulated regret of RAES and other baselines when $d = 10, 20$ against a $(1, 0)-$rational user with different $V_0$. RAES enjoys the same advantage as demonstrated in Figure 3. Since we have shown that the accumulated regret of RAES depends on $d$ quadratically, a larger time horizon $T$ is required to display its advantage for high-dimensional

problems. However, as $T$ becomes larger, the advantage of RAES also becomes more evident.

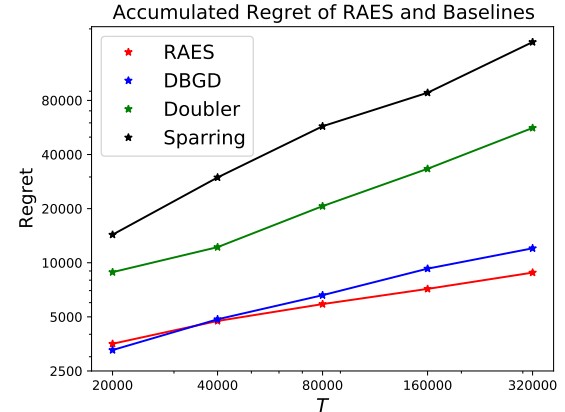 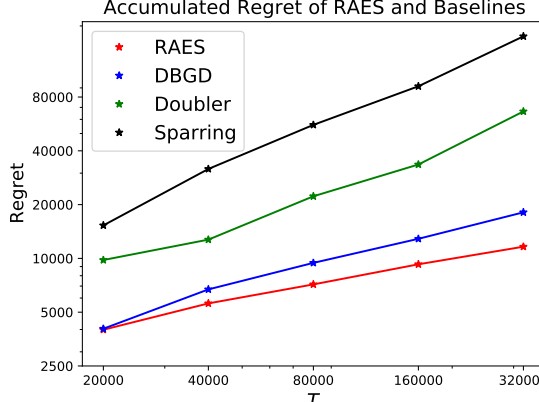

*Figure 5.* The accumulated regret of RAES and three baseline algorithms against a learning user with different $\gamma, V_0$, and $T$. Different colors specify different algorithms, and each star represents the accumulated regret (y-axis) of the algorithm given time horizon $T$ (x-axis) with $\gamma = 0.2$. Left: $V_0 = 100I_d$; right: $V_0$ is a diagonal matrix with half of its diagonal entries being 100 while the others being 1.

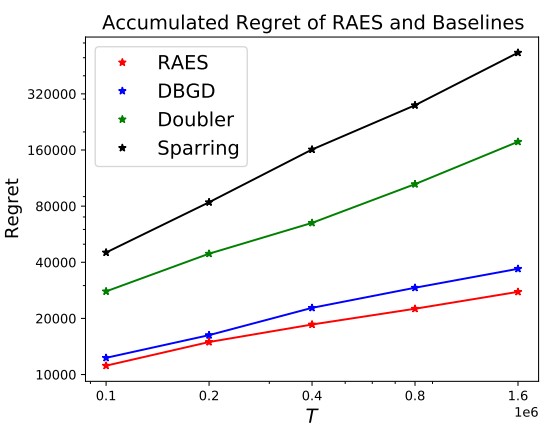 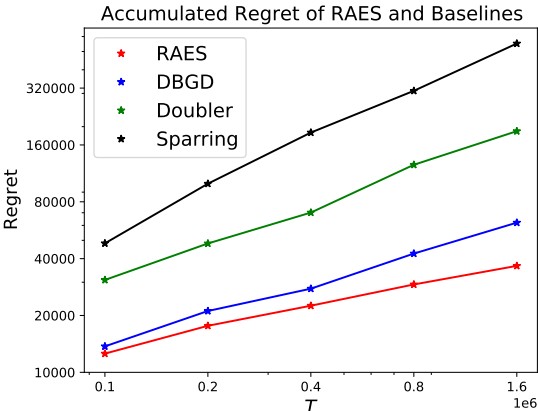

*Figure 6.* The accumulated regret of RAES and three baseline algorithms against a learning user with different $\gamma, V_0$, and $T$. Different colors specify different algorithms, and each star represents the accumulated regret (y-axis) of the algorithm given time horizon $T$ (x-axis) with $\gamma = 0.2$. Left: $V_0 = 100I_d$; right: $V_0$ is a diagonal matrix with half of its diagonal entries being 100 while the others being 1.