# OpenReview forum: "Learning from a Learning User for Optimal Recommendations"
_ICML.cc/2023/Workshop/ILHF — ILHF Workshop ICML 2023_

### Official Review · Reviewer_Jfy8 · 2023-06-07

**Rating:** 8
**Confidence:** 3

**Review:**

This paper studies an extension of the dueling bandit problem (comparative feedback between two items), where the user (and thus the utility function they're acting wrt) is non-stationary and dynamically evolves over time and is affected by historical interactions. A big part of the text is spent on walking through the proposed method, building up from a stylized version of the setup and explaining how the ellipsoid method is modified for this use case, together with the regret bounds - the writing is very clear which is appreciated, though I did not check the technical details regarding the theoretical proofs. Experiments were conducted on synthetic datasets and compared with several baselines to show the advantages of the proposed approach under various settings.

The learning environment studied by this work - "learning with learning users" - is an interesting framing and a departure from classical assumptions on stationary users and homogenous noise, and I think this more accurately reflects the reality. The proposed method is also an interesting adaptation of the ellipsoid method and I think the authors did a great job at explaining the approach.

---

### Official Review · Reviewer_3jxN · 2023-06-14

**Rating:** 8
**Confidence:** 3

**Review:**

This paper considers recommending content to a user who needs to explore different options to determine their value. They frame the problem as a dueling bandit / preference feedback problem where the user makes choices based on their estimated preference vector and an exploration term. The goal of the system is to eventually pick pairs of actions with high values under the ground-truth user reward function. They first consider learning with a perfect, static user (and derive an extension of the standard ellipsoid method). They then consider a learning user and derive an algorithm that recommends the user diverse content before eliciting accurate feedback from them.

The paper is very well written, stepping the reader through the key technical contributions in a sensible order. The experiments are somewhat simple but I think given more time, they could easily be improved.

---

### Decision · Program_Chairs · 2023-06-20

Accept